# General Planar Ideal Flow Solutions with No Symmetry Axis

**DOI:** 10.3390/ma16237378

**Published:** 2023-11-27

**Authors:** Sergei Alexandrov, Vyacheslav Mokryakov

**Affiliations:** 1Ishlinsky Institute for Problems in Mechanics RAS, 101-1 Prospect Vernadskogo, Moscow 119526, Russia; sergei_alexandrov@spartak.ru; 2Department of Civil Engineering, RUDN University, 6 Miklukho-Maklaya St., Moscow 117198, Russia

**Keywords:** ideal flows, double sliding and rotation model, Riemann’s method, plasticity, process design

## Abstract

Bulk ideal flows constitute a wide class of solutions in plasticity theory. Ideal flow solutions concern inverse problems. In particular, the solution determines part of the boundary of a region where it is valid. Bulk planar ideal flows exist in the case of (i) isotropic rigid/plastic material obeying an arbitrary pressure-independent yield criterion and its associated flow rule and (ii) the double sliding and rotation model based on the Mohr–Coulomb yield criterion. In the latter case, the intrinsic spin must vanish. Both models are perfectly plastic, and the complete equation systems are hyperbolic. All available specific solutions for both models describe flows with a symmetry axis. The present paper aims at general solutions for flows with no symmetry axis. The general structure of the solutions consists of two rigid regions connected by a plastic region. The characteristic lines between the plastic and rigid regions must be straight, which partly dictates the general structure of the characteristic nets. The solutions employ Riemann’s method in regions where the characteristics of both families are curvilinear. Special solutions that do not have such regions are considered separately. In any case, the solutions are practically analytical. A numerical technique is only necessary to evaluate ordinary integrals. The solutions found determine the tool shapes that produce ideal flows. In addition, the distribution of pressure over the tool’s surface is calculated, which is important for predicting the wear of tools.

## 1. Introduction

Ideal or streamlined flows constitute a class of solutions within the classical theory of rigid plasticity. The ideal flow condition is an additional equation within the standard system of equations of the latterly mentioned theory. Nevertheless, the resulting overdetermined system of equations is compatible, and restrictions are imposed on boundary conditions. Ideal flow solutions are design solutions. In particular, in the case of bulk stationary flows, a tool’s shape is not a given but is partly determined by the solution [1]. The first design solution of this kind has been presented in [2]. It was based on the method of characteristics. The traditional finite element method cannot calculate ideal flows. An ideal flow condition is advantageous for some deformation processes. In particular, ideal flows produce no redundant work [2]. This work is not zero in real processes. However, the experiment carried out in [3] has demonstrated that the ideal-flow die calculated in [2] is the best in terms of the following criteria: efficiency, uniformity of deformation, product strength, ductility, and fatigue life. This die has been compared to straight, convex circular, and concave circular die profiles. Therefore, ideal flow theory is used for the preliminary design of deformation processes [1]. This design aims to calculate the tool shape that produces an ideal flow. However, it has been seen from the solution in [2] that it is not unique. Therefore, other design criteria can be combined with the ideal flow condition. For example, the ideal flow solution for the axisymmetric drawing presented in [4] calculates the die of minimum length. This solution is for Tresca’s yield criterion. Papers [5,6] have demonstrated that steady and nonsteady three-dimensional ideal flows exist for the material model comprising this yield criterion and its associated flow rule. Only this material model has been associated with the concept of bulk ideal flows for a long time.

Many materials obey pressure-dependent yield criteria ([7,8,9,10] among many others). Various plastic flow rules are used in conjunction with such yield criteria. All these plastic flow rules can be conveniently divided into two groups. The models of one of these groups are plastically compressible. This property is inherent to many granular and similar materials. Models that account for this property have been proposed (in [11,12], among many others). However, in many cases, materials can be treated as plastically incompressible. The models of the other group describe the responses of these materials. Corresponding theories, solutions, and experiments have been reported in [13,14] for soils, granular materials [15,16], and traditional metals [17,18,19]. A test for verifying the theory proposed in [13] has been carried out in [20]. It has been concluded that the theory may be a viable law to describe the deformation of granular materials. The discrete element method has been employed in [21] to show that the model [22] for plastically incompressible materials provides a reasonable continuum mechanics description of the behavior of some granular materials. The present paper adopts the double shearing and rotation model proposed in [22]. Considering steady planar flows, it has been recently proven in [23] that the ideal flow condition is compatible with this material model if the intrinsic spin vanishes. Note that this condition is not satisfied for the model proposed in [13].

To the best of the authors’ knowledge, all available planar ideal flow solutions are symmetric relative to an axis. In particular, the solution [2] and its generalizations [24,25,26] are. On the other hand, some engineering applications require non-symmetric extrusion and drawing dies [27].

The present paper provides a steady planar ideal flow solution through a non-symmetric die. The theory developed in [23] is adopted. The solution for Tresca’s yield criterion is obtained as a particular case. Another particular case is the flow through a symmetric die. In these two particular cases, the new solution coincides with available solutions.

## 2. Statement of the Problem

A sheet of initial thickness *h*_in_ is extruded through a frictionless die. The final thickness of the sheet is *h*_out_. The process is stationary, and the deformation is plane strain. The elastic portion of the strain tensor is neglected. A plastic region in the die connects two rigid regions. One of the rigid regions moves horizontally with velocity *V*_out_ after exiting the plastic region. The other rigid region approaches the plastic region with velocity *V*_in_ at some angle *ψ*_in_ measured from the horizontal direction. A schematic diagram of the extrusion process is shown in Figure 1, where *F*_e_ is the extrusion force. The die shape that produces the above processes under the ideal flow condition should be found in the course of the solution.

It is assumed that the material obeys the double shearing and rotation model [22]. The constitutive equations are the Mohr–Coulomb yield criterion and the plastic flow rule. Let *σ_xx_*, *σ_yy_*, and *σ_xy_* be the stress components referred to Cartesian coordinates (*x*, *y*). The yield criterion is represented as
(1)$$\left(\sigma_{xx}+\sigma_{yy}\right)\sin\varphi +\sqrt{{\left(\sigma_{xx}-\sigma_{yy}\right)}^{2}+4\sigma_{xy}^{2}}=2k\cos\varphi ,$$
where *k* is the cohesion, and *φ* is the angle of internal friction. It is convenient to represent the plastic flow rule in terms of the angle of inclination of the direction of the algebraically greater principal stress to the *x*-axis. This angle is denoted as *ψ*. Then, the plastic flow rule becomes
(2)$$\begin{array}{l} \xi_{xx}+\xi_{yy}=0,\\ \sin2\psi \left(\xi_{xx}-\xi_{yy}\right)-2\cos2\psi \xi_{xy}-2\sin\varphi \left(\omega_{xy}+\Omega \right)=0. \end{array}$$

Here, $\xi_{xx}$, $\xi_{yy}$, and $\xi_{xy}$ are the strain rate components in the Cartesian coordinates; $\omega_{xy}$ is the only non-zero spin component in the Cartesian coordinates; and Ω is the intrinsic spin. In many cases, Ω=0 [22,28]. The present paper is concerned with this version of the model. Then, Equation (2) becomes
(3)$$\begin{array}{l} \xi_{xx}+\xi_{yy}=0,\\ \sin2\psi \left(\xi_{xx}-\xi_{yy}\right)-2\cos2\psi \xi_{xy}-2\sin\varphi \omega_{xy}=0. \end{array}$$

The constitutive equations above are supplemented with the stress equilibrium equations:(4)$$\frac{\partial \sigma_{xx}}{\partial x}+\frac{\partial \sigma_{xy}}{\partial y}=0\;\mathrm{and}\;\frac{\partial \sigma_{xy}}{\partial x}+\frac{\partial \sigma_{yy}}{\partial y}=0.$$

The ideal flow condition is that the trajectories of one of the principal stresses coincide with the streamlines. This condition and the equations above constitute an overdetermined system of equations. However, it has been shown in [23] that this system is compatible.

## 3. System of Equations in Characteristic Coordinates

Equations (1), (3), and (4) constitute a hyperbolic system. This section briefly summarizes available results concerning this system that are necessary for the subsequent solution.

Using a principal line coordinate system (*ξ*, *η*) for finding ideal flow solutions is convenient. The *ξ*-coordinate lines are the trajectories of the principal stress *σ_ξ_*, and the *η*-coordinate lines are the trajectories of the principal stress *σ_η_*. It is assumed without loss of generality that
(5)$$\sigma_{\xi }>\sigma_{\eta }.$$

Then, *ψ* involved in (3) is the anticlockwise angular rotation of the *ξ*-lines from the *x*-axis. The characteristic directions are determined as
(6)$$\frac{H_{\eta }}{H_{\xi }}\frac{d\eta }{d\xi }=-\tan\chi \;\mathrm{and}\;\frac{H_{\eta }}{H_{\xi }}\frac{d\eta }{d\xi }=\tan\chi ,$$
where *H_ξ_* and *H_η_* are the scale factors of the *ξ*- and *η*-coordinate curves, respectively, and χ=π/4+φ/2. The characteristic coordinates are denoted as (*α*, *β*). The first equation in (6) corresponds to the *α*-lines and the second to the *β*-lines. The ideal flow condition requires that the velocity vector **V** is tangent to the *ξ*-coordinate lines (Figure 2). The velocity vector may also be tangent to the *η*-coordinate lines. However, this condition is not required for the problem solved in the next section. The characteristic relations resulting from the stress equations are
(7)$$\begin{array}{l} \cos\varphi dp+2\left(p\sin\varphi +\cos\varphi \right)d\psi =0\;\mathrm{along}\;\mathrm{an}\;\alpha -\mathrm{line},\\ \cos\varphi dp-2\left(p\sin\varphi +\cos\varphi \right)d\psi =0\;\mathrm{along}\;a\;\beta -\mathrm{line}, \end{array}$$
where
(8)$$p=-\frac{\sigma_{\xi }+\sigma_{\eta }}{2k}.$$

It has been shown in [29] that it is always possible to put
(9)ψ=(α+β)cosφ
in regions where both characteristic families are curved. In this case, Equation (7) allows for *p* to be determined as
(10)$$\ln\left(\frac{p\sin\varphi +\cos\varphi }{p_{0}\sin\varphi +\cos\varphi }\right)=2\left(\beta -\alpha \right)\sin\varphi ,$$
where *p*_0_ is constant. It has been shown in [23] that the plastic flow rule is compatible with the ideal flow condition and the stress solution above if
(11)$$\frac{V}{V_{0}}={\left(\frac{p\sin\varphi +\cos\varphi }{p_{0}\sin\varphi +\cos\varphi }\right)}^{t},$$
where *V* is the magnitude of **V**, *V*_0_ is constant, and $t=\frac{\sin\varphi -1}{2\sin\varphi }$.

In regions where the *α*-lines are straight, Equations (9) and (10) are replaced with
(12)$$\psi =\left(\alpha_{0}+\beta \right)\cos\varphi ,\;\mathrm{and}\;\ln\left(\frac{p\sin\varphi +\cos\varphi }{p_{0}\sin\varphi +\cos\varphi }\right)=2\left(\beta -\alpha_{0}\right)\sin\varphi ,$$
where *α*_0_ is constant. Equation (11) is valid.

In regions where the *β*-lines are straight, Equations (9) and (10) are replaced with
(13)$$\psi =\left(\alpha +\beta_{0}\right)\cos\varphi \;\mathrm{and}\;\ln\left(\frac{p\sin\varphi +\cos\varphi }{p_{0}\sin\varphi +\cos\varphi }\right)=2\left(\beta_{0}-\alpha \right)\sin\varphi ,$$
where *β*_0_ is constant. Equation (11) is valid.

If the characteristic lines of both families are straight in a region, its motion is a rigid body translation.

The above equations show that calculating ideal flows mainly requires a characteristic network. The remaining calculations are merely manipulations with elementary functions.

## 4. Characteristic Network and Ideal Die Shapes

The solution for symmetric dies [23] allows for the general structure of the characteristic network to be guessed (Figure 3).

### 4.1. Special Solutions near the Die’s Exit

It is convenient to start analyzing this network from the die’s exit. Since the rigid region exiting the die moves horizontally, straight lines *AD* and *AD*’ are inclined to the horizontal line at angles *χ*, as shown in Figure 3. It has been taken into account here that the direction of the principal stress *σ_ξ_* is horizontal at point *A*. The position of point *A* relative to points *D* and *D*’ can be chosen arbitrarily, giving one of the parameters that control the characteristic network. It is convenient to choose the origin of the Cartesian coordinate system at point *A*. The distance between the *x*-axis and the center line of the sheet after exiting the die may serve as the noted parameter. This parameter is denoted as *â* (Figure 3). It varies in the range $-\frac{h_{\mathrm{out}}}{2}\leq \hat{a}\leq \frac{h_{\mathrm{out}}}{2}$. It follows from the geometry of Figure 3 that the lengths of lines *AD* and *AD*’ are
(14)$$r_{AD}=\left(\frac{h_{\mathrm{out}}}{2}-\hat{a}\right)\frac{1}{\sin\chi }\;\mathrm{and}\;r_{AD^{\prime }}=\left(\frac{h_{\mathrm{out}}}{2}+\hat{a}\right)\frac{1}{\sin\chi }.$$

Using this equation, one can find the Cartesian coordinates of points *D* and *D*’ in the form
(15)$$x_{D}=-\left(\frac{h_{\mathrm{out}}}{2}-\hat{a}\right)\cot\chi ,\;y_{D}=\frac{h_{\mathrm{out}}}{2}-\hat{a},\;x_{D^{\prime }}=-\left(\frac{h_{\mathrm{out}}}{2}+\hat{a}\right)\cot\chi ,\;\mathrm{and}\;y_{D^{\prime }}=-\left(\frac{h_{\mathrm{out}}}{2}+\hat{a}\right).$$

The orientation of the principal stress *σ_ξ_* at point *A* demands that *AD* is an *α*-line and *AD*’ is a *β*-line. Therefore, all *α*-lines are straight lines through *D* in region *AFD*. Similarly, all *β*-lines are straight lines through *D*’ in region *AF*’*D*’. It follows from (6) that the angle between the characteristic directions equals 2*χ*. Therefore, *AF* and *AF*’ are logarithmic spirals. In particular, *AF* can be represented as
(16)$$r=r_{AD}\exp\left(-\theta \tan\varphi \right).$$

Here, (*r*, *θ*) is the polar coordinate system with the origin at *D*, and *θ* is measured from line *AD* anticlockwise. Both families of characteristics are straight in *DFC*, where *DC* is a portion of the die’s surface. Therefore, the magnitude of angles *FDC* and *DCF* is *χ*. The inclination of *DC* to the *x*-axis is *γ*. The magnitude of this angle should be found in the course of the solution. It follows from the geometry of Figure 3 that the magnitude of angle *ADF* is also *γ*. Therefore, the length of *DF* is determined from (16) as
(17)$$r_{DF}=r_{AD}\exp\left(-\gamma \tan\varphi \right).$$

Using this equation, one can find the length of *DC* as
(18)$$r_{DC}=2r_{AD}\exp\left(-\gamma \tan\varphi \right)\cos\chi .$$

The Cartesian coordinates of point *C* are determined by employing (14), (15), and (18) in the form
(19)$$\begin{array}{l} x_{C}=\left(h_{out}-2\hat{a}\right)\cot\chi \left[\exp\left(-\gamma \tan\varphi \right)\cos\gamma -\frac{1}{2}\right],\\ y_{C}=\left(h_{out}-2\hat{a}\right)\left[\exp\left(-\gamma \tan\varphi \right)\cot\chi \sin\gamma +\frac{1}{2}\right]. \end{array}$$

Since the magnitude of angle *ADF* is *γ*, region *ADF* vanishes if *γ* = 0. Therefore, the solution under construction is valid if
(20)γ≥0.

Similarly, *AF*’ can be represented as
(21)$$r=r_{AD^{\prime }}\exp\left(\theta \tan\varphi \right).$$

Here, (*r*, *θ*) is the polar coordinate system with the origin at *D*’, and *θ* is measured from line *AD*’ clockwise. Both families of characteristics are straight in *D*’*F*’*C*’, where *D*’*C*’ is a portion of the die’s surface. Therefore, the magnitude of angles *F*’*D*’*C*’ and *D*’*C*’*F*’ is *χ*. The inclination of *D*’*C*’ to the *x*-axis is *γ*’. The magnitude of this angle should be found in the course of the solution. It follows from the geometry of Figure 3 that the magnitude of angle *AD’F*’ is also *γ*’. Therefore, the length of *D*’*F*’ is determined from (21) as
(22)$$r_{D^{\prime }F^{\prime }}=r_{AD^{\prime }}\exp\left(-\gamma^{\prime }\tan\varphi \right).$$

Using this equation, one can find the length of *D*’*C*’ as
(23)$$r_{D^{\prime }C^{\prime }}=2r_{AD^{\prime }}\exp\left(-\gamma^{\prime }\tan\varphi \right)\cos\chi .$$

The Cartesian coordinates of point *C*’ are determined by employing (14), (15), and (23) in the form
(24)$$\begin{array}{l} x_{C^{\prime }}=\left(h_{out}+2\hat{a}\right)\cot\chi \left[\exp\left(-\gamma \prime \tan\varphi \right)\cos\gamma^{\prime }-\frac{1}{2}\right],\\ y_{C^{\prime }}=-\left(h_{out}+2\hat{a}\right)\left[\exp\left(-\gamma^{\prime }\tan\varphi \right)\cot\chi \sin\gamma^{\prime }+\frac{1}{2}\right]. \end{array}$$

Since the magnitude of angle *AD*’*F*’ is *γ*’, region *AD*’*F*’ vanishes if *γ*’ = 0. Therefore, the solution under construction is valid if
(25)$$\gamma^{\prime }\geq 0.$$

### 4.2. Region AFBF’ (Figure 3)

The characteristics of both families are curved in this region. Since *ψ* = 0 at point *A*, Equation (9) demands that the base *α*-line coincides with *AF*’ and the base *β*-line with *AF* (i.e., *α* = 0 on line *AF*, and *β* = 0 on line *AF*’). It follows from the geometry of Figure 3 that *ψ* = *γ* at point *F* and *ψ* = *γ*’ at point *F*’. Therefore, Equation (9) supplies the characteristic coordinates of points *F*’ and *F* in the form
(26)$$\alpha_{F^{\prime }}=-\frac{\gamma^{\prime }}{\cos\varphi }\;\mathrm{and}\;\beta_{F}=\frac{\gamma }{\cos\varphi }.$$

The radii of curvature of the *α*- and *β*-characteristic lines are denoted as *R* and *S*, respectively. These quantities can be defined as
(27)$$\frac{1}{R}=\frac{\partial \psi }{\partial s_{\alpha }}\;\mathrm{and}\;\frac{1}{S}=-\frac{\partial \psi }{\partial s_{\beta }},$$
where $\frac{\partial }{\partial s_{\alpha }}$ and $\frac{\partial }{\partial s_{\beta }}$ denote differentiation along the *α*- and *β*-characteristic lines, respectively. It is advantageous to introduce the following quantities:(28)$$R_{0}=R\exp\left[\left(\beta -\alpha \right)\sin\varphi \right]\;\mathrm{and}\;S_{0}=S\exp\left[\left(\beta -\alpha \right)\sin\varphi \right].$$

It has been shown in [29] that these quantities satisfy the equations:(29)$$\frac{\partial R_{0}}{\partial \beta }=S_{0}\;\mathrm{and}\;\frac{\partial S_{0}}{\partial \alpha }=-R_{0}.$$

One can transform these equations to
(30)$$\frac{\partial^{2}R_{0}}{\partial \alpha \partial \beta }+R_{0}=0\;\mathrm{and}\;\frac{\partial^{2}S_{0}}{\partial \alpha \partial \beta }+S_{0}=0.$$

Each of these equations is the equation of telegraphy. Therefore, they can be integrated by the method of Riemann. In particular, the Green’s function is
(31)$$G\left(a,\;b,\;\alpha ,\;\beta \right)=J_{0}\left[2\sqrt{\left(a-\alpha \right)\left(b-\beta \right)}\right],$$
where $J_{0}\left(z\right)$ is the Bessel function of zero order.

Formulating the boundary conditions on lines *AF* and *AF*’ requires calculating the radii of curvature of these lines. Differential geometry supplies the formula for the radius of curvature of a curve represented in polar coordinates. Substituting (16) and (21) into this formula and using (14), one arrives at
(32)$$S_{AF}=-\frac{\left(h_{\mathrm{out}}-2\hat{a}\right)}{2\sin\chi \cos\varphi }\exp\left(-\theta \tan\varphi \right)\;\mathrm{and}\;R_{AF^{\prime }}=-\frac{\left(h_{\mathrm{out}}+2\hat{a}\right)}{2\sin\chi \cos\varphi }\exp\left(\theta \tan\varphi \right),$$
where *S_AF_* is the value of *S* on line *AF*, and *R_AF’_* is the value of *R* on line *AF*’. Since *ψ* = 0, and *θ* = 0 at point *A*, Equations (9) and (32) combine to give
(33)$$S_{AF}\left(\beta \right)=-\frac{\left(h_{out}-2\hat{a}\right)}{2\sin\chi \cos\varphi }\exp\left(-\beta \sin\varphi \right)\;\mathrm{and}\;R_{AF^{\prime }}\left(\alpha \right)=-\frac{\left(h_{out}+2\hat{a}\right)}{2\sin\chi \cos\varphi }\exp\left(\alpha \sin\varphi \right).$$

Substituting *S_AF_*(*β*) into the second equation in (28) at *α* = 0 and *R_AF’_*(*α*) into the first equation in (28) at *β* = 0 supplies
(34)$$S_{0}^{AF}=-\frac{\left(h_{out}-2\hat{a}\right)}{2\sin\chi \cos\varphi }\;\mathrm{and}\;R_{0}^{AF^{\prime }}=-\frac{\left(h_{out}+2\hat{a}\right)}{2\sin\chi \cos\varphi },$$
where $S_{0}^{AF}$ is the value of *S*_0_ on line *AF,* and $R_{0}^{AF^{\prime }}$ is the value *R*_0_ on line *AF*’. The equations in (34) are the boundary conditions for the equations in (30).

According to Riemann’s method, the value of *S*_0_ at a generic point *P* in region *AFBF*’ is determined from the following equation (Figure 4):(35)$$\begin{array}{l} \int_{PP_{\alpha }}\left(S_{0}\frac{\partial G}{\partial \beta }-G\frac{\partial S_{0}}{\partial \beta }\right)d\beta +\int_{P_{\alpha }A}\left(G\frac{\partial S_{0}}{\partial \alpha }-S_{0}\frac{\partial G}{\partial \alpha }\right)d\alpha +\\ \;+\int_{AP_{\beta }}\left(S_{0}\frac{\partial G}{\partial \beta }-G\frac{\partial S_{0}}{\partial \beta }\right)d\beta +\int_{P_{\beta }P}\left(G\frac{\partial S_{0}}{\partial \alpha }-S_{0}\frac{\partial G}{\partial \alpha }\right)d\alpha =0. \end{array}$$

These integrals are evaluated separately below. The value of *S*_0_ at point *P* is denoted as $S_{0}^{P}$.

It follows from (31) that *G* = 1 on lines *P_β_P* and *PP_α_*. Therefore,
(36)$$\int_{PP_{\alpha }}\left(S_{0}\frac{\partial G}{\partial \beta }-G\frac{\partial S_{0}}{\partial \beta }\right)d\beta =-S_{0}^{AF\prime }+S_{0}^{P}\;\mathrm{and}\;\int_{P_{\beta }P}\left(G\frac{\partial S_{0}}{\partial \alpha }-S_{0}\frac{\partial G}{\partial \alpha }\right)d\alpha =S_{0}^{P}-S_{0}^{AF},$$
where $S_{0}^{AF^{\prime }}$ is the value of *S*_0_ on line *AF*’. Employing (34), one can integrate the second equation in (29) to get
(37)$$S_{0}^{AF^{\prime }}=-R_{0}^{AF^{\prime }}\alpha +S_{0}^{AF}.$$

Since $S_{0}=S_{0}^{AF}$ on *AP_β_* and *G* = 1 at point *P_β_*,
(38)$$\int_{AP_{\beta }}\left(S_{0}\frac{\partial G}{\partial \beta }-G\frac{\partial S_{0}}{\partial \beta }\right)d\beta =S_{0}^{AF}\left[1-J_{0}\left(2\sqrt{ab}\right)\right].$$

Here, Equation (31) has been used. Integrating by parts, one can transform the second integral in (35) as
(39)$$\begin{array}{l} \int_{P_{\alpha }A}\left(G\frac{\partial S_{0}}{\partial \alpha }-S_{0}\frac{\partial G}{\partial \alpha }\right)d\alpha =\int_{a}^{0}\left(G\frac{\partial S_{0}}{\partial \alpha }-S_{0}\frac{\partial G}{\partial \alpha }\right)d\alpha =\\ \;=\int_{a}^{0}\left[2G\frac{\partial S_{0}}{\partial \alpha }-\frac{\partial \left(S_{0}G\right)}{\partial \alpha }\right]d\alpha =2\int_{a}^{0}G\frac{\partial S_{0}}{\partial \alpha }d\alpha -{\left(S_{0}G\right)|}_{\alpha =0}+{\left(S_{0}G\right)|}_{\alpha =a}. \end{array}$$

Since $S_{0}=S_{0}^{AF^{\prime }}$ on *AP_α_*, substituting (30) and (36) into (38) leads to
(40)$$\int_{P_{\alpha }A}\left(G\frac{\partial S_{0}}{\partial \alpha }-S_{0}\frac{\partial G}{\partial \alpha }\right)d\alpha =S_{0}^{AF}\left[1-J_{0}\left(2\sqrt{ab}\right)\right]-2R_{0}^{AF\prime }\int_{a}^{0}J_{0}\left[2\sqrt{\left(a-\alpha \right)b}\right]d\alpha .$$

This result can be further simplified taking into account that
(41)$$d\left[zJ_{1}\left(z\right)\right]/dz=zJ_{0}\left(z\right),$$
where $J_{1}\left(z\right)$ is the Bessel function of first order. Put $z=2\sqrt{\left(a-\alpha \right)b}$. Then, using (41), one gets
(42)$$J_{0}\left[2\sqrt{\left(a-\alpha \right)b}\right]=-\frac{d\left\{\sqrt{\left(a-\alpha \right)b}J_{1}\left[2\sqrt{\left(a-\alpha \right)b}\right]\right\}}{bd\alpha }.$$

Substituting (42) into (40) yields
(43)$$\int_{P_{\alpha }A}\left(G\frac{\partial S_{0}}{\partial \alpha }-S_{0}\frac{\partial G}{\partial \alpha }\right)d\alpha =S_{0}^{AF}\left[1-J_{0}\left(2\sqrt{ab}\right)\right]+\frac{2\sqrt{ab}R_{0}^{AF^{\prime }}}{b}J_{1}\left(2\sqrt{ab}\right).$$

Equations (35), (36), (38), and (43) combine to give
(44)$$S_{0}^{P}=S_{0}^{AF}J_{0}\left(2\sqrt{ab}\right)-\frac{\sqrt{ab}R_{0}^{AF^{\prime }}}{b}J_{1}\left(2\sqrt{ab}\right).$$

One can eliminate $S_{0}^{AF}$ and $R_{0}^{AF\prime }$ in this equation employing (34). It is seen from (26) that *ab* < 0. Therefore, it is convenient to rewrite (44) as
(45)$$S_{0}^{P}=S_{0}^{AF}I_{0}\left(2\sqrt{\left|ab\right|}\right)+\sqrt{\left|\frac{a}{b}\right|}R_{0}^{AF^{\prime }}I_{1}\left(2\sqrt{\left|ab\right|}\right).$$

Here, $I_{0}\left(z\right)$ and $I_{1}\left(z\right)$ are the modified Bessel functions of zero and first orders, respectively.

Using the same line of reasoning as above, one can represent the solution of the first equation in (30) as
(46)$$R_{0}^{P}=R_{0}^{AF^{\prime }}I_{0}\left(2\sqrt{\left|ab\right|}\right)+S_{0}^{AF}\sqrt{\left|\frac{b}{a}\right|}I_{1}\left(2\sqrt{\left|ab\right|}\right).$$

The radius of curvature of line *FB* is determined from (28) and (46) as
(47)$$R_{FB}\left(\alpha \right)=\left[R_{0}^{AF^{\prime }}I_{0}\left(2\sqrt{\left|\alpha b_{F}\right|}\right)+S_{0}^{AF}\sqrt{\left|\frac{b_{F}}{\alpha }\right|}I_{1}\left(2\sqrt{\left|\alpha b_{F}\right|}\right)\right]\exp\left[\left(\alpha -\beta_{F}\right)\sin\varphi \right].$$

Therefore, the Cartesian coordinates of this line can be represented by the equations
(48)$$\begin{array}{l} x_{FB}=x_{F}+\cos\varphi \int_{0}^{\alpha }\left[R_{0}^{AF^{\prime }}I_{0}\left(2\sqrt{\left|ab_{F}\right|}\right)+S_{0}^{AF}\sqrt{\left|\frac{b_{F}}{a}\right|}I_{1}\left(2\sqrt{\left|ab_{F}\right|}\right)\right]\exp\left[\left(a-\beta_{F}\right)\sin\varphi \right]\cos\left(\psi -\chi \right)da,\\ y_{FB}=y_{F}+\cos\varphi \int_{0}^{\alpha }\left[R_{0}^{AF^{\prime }}I_{0}\left(2\sqrt{\left|ab_{F}\right|}\right)+S_{0}^{AF}\sqrt{\left|\frac{b_{F}}{a}\right|}I_{1}\left(2\sqrt{\left|ab_{F}\right|}\right)\right]\exp\left[\left(a-\beta_{F}\right)\sin\varphi \right]\sin\left(\psi -\chi \right)da. \end{array}$$

Here *ψ* is determined from (9) as *ψ =* (*a + β_F_*) cos *φ* and $\alpha_{F^{\prime }}\leq a\leq 0$. The *x*- and *y*- coordinates of point *F* are readily found from (14) and (17) as
(49)$$\begin{array}{l} x_{F}=r_{DF}\cos\left(\chi -\gamma \right)-r_{AD}\cos\chi =\left(\frac{h_{\mathrm{out}}}{2}-\hat{a}\right)\left[\frac{\cos\left(\chi -\gamma \right)}{\sin\chi }\exp\left(-\gamma \tan\varphi \right)-\cot\chi \right],\\ y_{F}=\frac{h_{\mathrm{out}}}{2}-r_{DF}\sin\left(\chi -\gamma \right)=\left(\frac{h_{\mathrm{out}}}{2}-\hat{a}\right)\left[1-\frac{\exp\left(-\gamma \tan\varphi \right)\sin\left(\chi -\gamma \right)}{\sin\chi }\right]. \end{array}$$

The radius of curvature of line *F*’*B* is determined from (28) and (45) as
(50)$$S_{F^{\prime }B}\left(\beta \right)=\left[S_{0}^{AF}I_{0}\left(2\sqrt{\left|a_{F^{\prime }}\beta \right|}\right)+\sqrt{\left|\frac{a_{F\prime }}{\beta }\right|}R_{0}^{AF^{\prime }}I_{1}\left(2\sqrt{\left|a_{F^{\prime }}\beta \right|}\right)\right]\exp\left[\left(\alpha_{F^{\prime }}-\beta \right)\sin\varphi \right].$$

Therefore, the Cartesian coordinates of this line can be represented by the equations
(51)$$\begin{array}{l} x_{F^{\prime }B}=x_{F^{\prime }}-\cos\varphi \int_{0}^{\beta }\left[S_{0}^{AF}I_{0}\left(2\sqrt{\left|a_{F^{\prime }}b\right|}\right)+\sqrt{\left|\frac{a_{F^{\prime }}}{b}\right|}R_{0}^{AF^{\prime }}I_{1}\left(2\sqrt{\left|a_{F^{\prime }}b\right|}\right)\right]\exp\left[\left(\alpha_{F^{\prime }}-b\right)\sin\varphi \right]\cos\left(\psi +\chi \right)db,\\ y_{F^{\prime }B}=y_{F^{\prime }}-\cos\varphi \int_{0}^{\beta }\left[S_{0}^{AF}I_{0}\left(2\sqrt{\left|a_{F^{\prime }}b\right|}\right)+\sqrt{\left|\frac{a_{F^{\prime }}}{b}\right|}R_{0}^{AF^{\prime }}I_{1}\left(2\sqrt{\left|a_{F^{\prime }}b\right|}\right)\right]\exp\left[\left(\alpha_{F^{\prime }}-b\right)\sin\varphi \right]\sin\left(\psi +\chi \right)db. \end{array}$$

Here *ψ* is determined from (9) as *ψ =* (*α_F’_ + b*) cos *φ* and $0\leq b\leq \beta_{F}$. The *x*- and *y*- coordinates of point *F*’ are readily found from (14) and (22) as
(52)$$\begin{array}{l} x_{F^{\prime }}=r_{D^{\prime }F^{\prime }}\cos\left(\chi -\gamma^{\prime }\right)-r_{AD^{\prime }}\cos\chi =\left(\frac{h_{\mathrm{out}}}{2}+\hat{a}\right)\left[\frac{\cos\left(\chi -\gamma^{\prime }\right)}{\sin\chi }\exp\left(-\gamma^{\prime }\tan\varphi \right)-\cot\chi \right],\\ y_{F^{\prime }}=\frac{h_{\mathrm{out}}}{2}-r_{DF}\sin\left(\chi -\gamma \right)=\left(\frac{h_{\mathrm{out}}}{2}+\hat{a}\right)\left[1-\frac{\exp\left(-\gamma^{\prime }\tan\varphi \right)\sin\left(\chi -\gamma^{\prime }\right)}{\sin\chi }\right]. \end{array}$$

The Cartesian coordinates of point *B*, *x_B_* and *y_B_*, are determined from (48) at *a* = *α_F’_* or (51) at *b* = *β_F_*.

### 4.3. Regions FCEB and F’C’E’B (Figure 3)

The *β*-lines are straight in region *FCEB*. Therefore, the angle *ψ* is independent of *β*. Since *ψ* = *γ* and *α* = 0 on line *CF*, the dependence of this angle on *α* follows from (9) in the form
(53)ψ=αcosφ+γ.

It follows from the geometry of Figure 3 that
(54)$$\begin{array}{l} \frac{\partial x}{\partial \alpha }=R\cos\varphi \cos\left(\psi -\frac{\pi }{4}-\frac{\varphi }{2}\right),\;\frac{\partial x}{\partial \beta }=-T\left(\alpha \right)\sin\left(\psi -\frac{\pi }{4}+\frac{\varphi }{2}\right),\\ \frac{\partial y}{\partial \alpha }=R\cos\varphi \sin\left(\psi -\frac{\pi }{4}-\frac{\varphi }{2}\right),\;\frac{\partial y}{\partial \beta }=T\left(\alpha \right)\cos\left(\psi -\frac{\pi }{4}+\frac{\varphi }{2}\right). \end{array}$$

Here *T*(*α*) is an arbitrary function of *α*. The compatibility equations are
(55)$$\frac{\partial^{2}x}{\partial \alpha \partial \beta }=\frac{\partial^{2}x}{\partial \beta \partial \alpha }\;\mathrm{and}\;\frac{\partial^{2}y}{\partial \alpha \partial \beta }=\frac{\partial^{2}y}{\partial \beta \partial \alpha }.$$

Substituting (53) and (54) into (55) yields
(56)$$\begin{array}{l} \frac{\partial R}{\partial \beta }\cos\varphi \cos\left(\psi -\chi \right)=\frac{dT}{d\alpha }\cos\left(\psi +\chi \right)-T\sin\left(\psi +\chi \right)\cos\varphi ,\\ \frac{\partial R}{\partial \beta }\cos\varphi \sin\left(\psi -\chi \right)=\frac{dT}{d\alpha }\sin\left(\psi +\chi \right)+T\cos\left(\psi +\chi \right)\cos\varphi . \end{array}$$

One can solve these equations for the derivatives dT/dα and ∂R/∂β. As a result,
(57)$$\frac{dT}{d\alpha }=T\sin\varphi \;\mathrm{and}\;\frac{\partial R}{\partial \beta }=-\frac{T}{\cos\varphi }.$$

The solution of the first equation satisfying the condition *T* = 1 for *α =* 0 is
(58)$$T=e^{\alpha \sin\varphi }.$$

Substituting this solution in the second equation in (57) and integrating the resulting equation leads to
(59)$$R=-\frac{e^{\alpha \sin\varphi }}{\cos\varphi }\left(\beta -\beta_{F}\right)+R_{FB}\left(\alpha \right).$$

Here, the last term can be eliminated by employing (45), giving the dependence of *R* on *α* and *β* in region *FCEB*.

Curve *CE* bisects the angle between the *α*- and *β*- directions. Therefore, the equation for this curve is
(60)$$ds_{\alpha }=ds_{\beta }.$$

It follows from (56) that *ds_α_* = *R* cos *φ dα*, and *ds_β_* = *Tdβ*. These Equations (58), (59), and (60) combine to give
(61)$$\frac{d\beta }{d\alpha }=\cos\varphi R_{FB}\left(\alpha \right)e^{-\alpha \sin\varphi }-\beta +\beta_{F}.$$

The solution of this equation satisfying the boundary condition *β* = *β_C_* for *α* = 0 is
(62)$$\beta =\beta_{F}+\left(\beta_{C}-\beta_{F}\right)e^{-\alpha }+e^{-\alpha }\cos\varphi \int_{0}^{\alpha }R_{FB}\left(\mu \right)e^{\mu \left(1-\sin\varphi \right)}d\mu .$$

Using (59) and (60), one can derive the following equations for determining curve *CE*:(63)$$\begin{array}{l} \frac{dx}{d\alpha }=2\cos\chi \cos\psi \left[R_{FB}\left(\alpha \right)\cos\varphi -e^{\alpha \sin\varphi }\left(\beta -\beta_{F}\right)\right],\\ \frac{dy}{d\alpha }=2\cos\chi \sin\psi \left[R_{FB}\left(\alpha \right)\cos\varphi -e^{\alpha \sin\varphi }\left(\beta -\beta_{F}\right)\right]. \end{array}$$

In these equations, *β* should be eliminated by means of (62) and *ψ* by means of (53). Integrating the resulting equations gives
(64)$$\begin{array}{l} x_{CE}=2\cos\chi \int_{0}^{\alpha }\cos\psi \left[R_{FB}\left(\eta \right)\cos\varphi -e^{\eta \sin\varphi }\left(\beta -\beta_{F}\right)\right]d\eta +x_{C},\\ y_{CE}=2\cos\chi \int_{0}^{\alpha }\sin\psi \left[R_{FB}\left(\eta \right)\cos\varphi -e^{\eta \sin\varphi }\left(\beta -\beta_{F}\right)\right]d\eta +y_{C}. \end{array}$$

Here $0\leq \alpha \leq \alpha_{F^{\prime }}$. Equation (19) provides the values of *x_C_* and *y_C_*. Integrating by parts in (64) simplifies subsequent numerical integration. As a result,
(65)$$\begin{array}{l} x_{C^{\prime }E^{\prime }}\left(\alpha \right)=2\cos\chi \cos\varphi \int_{0}^{\alpha }R_{FB}\left(\mu \right)\cos\left(\mu \cos\varphi +\gamma \right)d\mu -\\ \;\;-\frac{\left(\beta_{C}-\beta_{F}\right)\cos\chi }{\left(1-\sin\varphi \right)}\left\{v_{1}\left(\alpha \right)e^{-\alpha \left(1-\sin\varphi \right)}+\cos\gamma -\sin\left(\varphi +\gamma \right)\right\}-\\ \;\;-\frac{\cos\chi \cos\varphi }{\left(1-\sin\varphi \right)}\left\{v_{1}\left(\alpha \right)e^{-\alpha \left(1-\sin\varphi \right)}\int_{0}^{\alpha }R_{FB}\left(\mu \right)e^{\mu \left(1-\sin\varphi \right)}d\mu -\int_{0}^{\alpha }R_{F^{\prime }B}\left(\mu \right)v_{1}\left(\mu \right)d\mu \right\}+x_{C},\\ y_{C^{\prime }E^{\prime }}\left(\beta \right)=2\cos\chi \cos\varphi \int_{0}^{\alpha }R_{FB}\left(\mu \right)\sin\left(\mu \cos\varphi +\gamma \right)d\mu +\\ \;\;\;\;\;+\frac{\left(\beta_{C}-\beta_{F}\right)\cos\chi }{\left(1-\sin\varphi \right)}\left[v_{2}\left(\alpha \right)e^{-\alpha \left(1-\sin\varphi \right)}-\sin\gamma -\cos\left(\varphi +\gamma \right)\right]+\\ \;\;+\frac{\cos\chi \cos\varphi }{\left(1-\sin\varphi \right)}\left\{v_{2}\left(\alpha \right)e^{-\alpha \left(1-\sin\varphi \right)}\int_{0}^{\alpha }R_{FB}\left(\mu \right)e^{\mu \left(1-\sin\varphi \right)}d\mu -\int_{0}^{\alpha }R_{FB}\left(\mu \right)v_{2}\left(\mu \right)d\mu \right\}+y_{C}, \end{array}$$
where
(66)$$\begin{array}{l} v_{1}\left(z\right)=\sin\left(z\cos\varphi +\gamma +\varphi \right)-\cos\left(z\cos\varphi +\gamma \right),\\ v_{2}\left(z\right)=\sin\left(z\cos\varphi +\gamma \right)+\cos\left(z\cos\varphi +\gamma +\varphi \right). \end{array}$$

This solution supplies the shape of wall *CE* in parametric form. In particular, the Cartesian coordinates of point *E* are determined from (65) as
(67)$$x_{E}=x_{CE}\left(\alpha_{F^{\prime }}\right)\;\mathrm{and}\;y_{E}=y_{CE}\left(\alpha_{F^{\prime }}\right).$$

Region *F*’*C*’*E*’*B*, where the *α*-lines are straight, can be treated similarly. The angle *ψ* is independent of *α*. Since *ψ* = −*γ*’ and *α* = 0 on line *C*’*F*’, the dependence of this angle on *β* follows from (9) in the form
(68)$$\psi =\beta \cos\varphi -\gamma^{\prime }.$$

It follows from the geometry of Figure 3 that
(69)$$\begin{array}{l} \frac{\partial x}{\partial \alpha }=T_{2}\left(\beta \right)\cos\left(\psi -\frac{\pi }{4}-\frac{\varphi }{2}\right),\;\frac{\partial x}{\partial \beta }=S\cos\varphi \sin\left(\psi -\frac{\pi }{4}+\frac{\varphi }{2}\right),\\ \frac{\partial y}{\partial \alpha }=T_{2}\left(\beta \right)\sin\left(\psi -\frac{\pi }{4}-\frac{\varphi }{2}\right),\;\frac{\partial y}{\partial \beta }=-S\cos\varphi \cos\left(\psi -\frac{\pi }{4}+\frac{\varphi }{2}\right). \end{array}$$

Here, *T*_2_(*β*) is an arbitrary function of *β*. Substituting (68) and (69) into (55) yields
(70)$$\begin{array}{l} \frac{\partial S}{\partial \alpha }\cos\varphi \cos\left(\psi +\chi \right)=-\frac{dT_{2}}{d\beta }\cos\left(\psi -\chi \right)+T_{2}\sin\left(\psi -\chi \right)\cos\varphi ,\\ \frac{\partial S}{\partial \alpha }\cos\varphi \sin\left(\psi +\chi \right)=-\frac{dT_{2}}{d\beta }\sin\left(\psi -\chi \right)-T_{2}\cos\left(\psi -\chi \right)\cos\varphi . \end{array}$$

One can solve these equations for the derivatives $\frac{dT_{2}}{d\beta }$ and $\frac{\partial S}{\partial \alpha }$. As a result,
(71)$$\frac{dT_{2}}{d\beta }=-T_{2}\sin\varphi ,\;\mathrm{and}\;\frac{\partial S}{\partial \alpha }=-\frac{T_{2}}{\cos\varphi }.$$

The solution of the first equation satisfying the condition *T*_2_ = 1 for *β* = 0 is
(72)$$T_{2}=e^{-\beta \sin\varphi }.$$

Substituting this solution in the second equation in (71) and integrating the resulting equation leads to
(73)$$S=-\frac{\exp\left(-\beta \sin\varphi \right)}{\cos\varphi }\left(\alpha -\alpha_{F^{\prime }}\right)+S_{F^{\prime }B}\left(\beta \right).$$

Here, the last term can be eliminated by employing (50), giving the dependence of *S* on *α* and *β* in region *F*’*C*’*E*’*B*.

Curve *C*’*E*’ bisects the angle between the *α*- and *β*- directions. Therefore, Equation (60) is valid. It follows from (70) that $ds_{\beta }=-S\cos\varphi d\beta$, and $ds_{\alpha }=T_{2}d\alpha$. These Equations, (60), (72), and (73) combine to give
(74)$$\frac{d\alpha }{d\beta }=-\cos\varphi S_{F^{\prime }B}\left(\beta \right)e^{\beta \sin\varphi }+\alpha -\alpha_{F^{\prime }}.$$

The solution of this equation satisfying the boundary condition *α* = *α_C’_* for *β* = 0 is
(75)$$\alpha =\alpha_{F^{\prime }}+\left(\alpha_{C^{\prime }}-\alpha_{F^{\prime }}\right)e^{\beta }-e^{\beta }\cos\varphi \int_{0}^{\beta }S_{F^{\prime }B}\left(\mu \right)e^{-\mu \left(1-\sin\varphi \right)}d\mu .$$

Using (60) and (73), one can derive the equations for determining curve *C*’*E*’ in the form
(76)$$\begin{array}{l} \frac{dx}{d\beta }=-2\cos\chi \cos\psi \left[S_{F^{\prime }B}\left(\beta \right)\cos\varphi -e^{-\beta \sin\varphi }\left(\alpha -\alpha_{F^{\prime }}\right)\right],\\ \frac{dy}{d\beta }=-2\cos\chi \sin\psi \left[S_{F^{\prime }B}\left(\beta \right)\cos\varphi -e^{-\beta \sin\varphi }\left(\alpha -\alpha_{F^{\prime }}\right)\right]. \end{array}$$

In these equations, *α* should be eliminated by means of (75) and *ψ* by means of (68). Integrating the resulting equations gives
(77)$$\begin{array}{l} x_{C^{\prime }E^{\prime }}\left(\beta \right)=-2\cos\chi \int_{0}^{\beta }\cos\psi \left[S_{F^{\prime }B}\left(\mu \right)\cos\varphi -\exp^{-\mu \sin\varphi }\left(\alpha -\alpha_{F^{\prime }}\right)\right]d\mu +x_{C^{\prime }},\\ y_{C^{\prime }E^{\prime }}\left(\beta \right)=-2\cos\chi \int_{0}^{\beta }\sin\psi \left[S_{F^{\prime }B}\left(\mu \right)\cos\varphi -\exp^{-\mu \sin\varphi }\left(\alpha -\alpha_{F^{\prime }}\right)\right]d\mu +y_{C^{\prime }}. \end{array}$$

Here $0\leq \beta \leq \beta_{F}$. Equation (19) provides the values of *x_C’_* and *y_C’_*. Integrating by parts in (77) simplifies subsequent numerical integration. As a result,
(78)$$\begin{array}{l} x_{C^{\prime }E^{\prime }}\left(\beta \right)=-2\cos\chi \cos\varphi \int_{0}^{\beta }S_{F^{\prime }B}\left(\mu \right)\cos\left(\mu \cos\varphi -\gamma^{\prime }\right)d\mu +\\ \;\;+\frac{\left(\alpha_{C^{\prime }}-\alpha_{F^{\prime }}\right)\cos\chi }{\left(1-\sin\varphi \right)}\left\{u_{1}\left(\beta \right)e^{\beta \left(1-\sin\varphi \right)}-\cos\gamma^{\prime }+\sin\left(\varphi +\gamma^{\prime }\right)\right\}-\\ \;\;-\frac{\cos\chi \cos\varphi }{\left(1-\sin\varphi \right)}\left\{u_{1}\left(\beta \right)e^{\beta \left(1-\sin\varphi \right)}\int_{0}^{\beta }S_{F^{\prime }B}\left(\mu \right)e^{-\mu \left(1-\sin\varphi \right)}d\mu -\int_{0}^{\beta }S_{F^{\prime }B}\left(\mu \right)u_{1}\left(\mu \right)d\mu \right\},\\ y_{C^{\prime }E^{\prime }}\left(\beta \right)=-2\cos\chi \cos\varphi \int_{0}^{\beta }S_{F^{\prime }B}\left(\mu \right)\sin\left(\mu \cos\varphi -\gamma^{\prime }\right)d\mu +\\ \;+\frac{\left(\alpha_{C^{\prime }}-\alpha_{F^{\prime }}\right)\cos\chi }{\left(1-\sin\varphi \right)}\left[u_{2}\left(\beta \right)e^{\beta \left(1-\sin\varphi \right)}+\sin\gamma^{\prime }+\cos\left(\varphi +\gamma^{\prime }\right)\right]-\\ \;\;-\frac{\cos\chi \cos\varphi }{\left(1-\sin\varphi \right)}\left\{u_{2}\left(\beta \right)e^{\beta \left(1-\sin\varphi \right)}\int_{0}^{\beta }S_{F^{\prime }B}\left(\mu \right)e^{-\mu \left(1-\sin\varphi \right)}d\mu -\int_{0}^{\beta }S_{F^{\prime }B}\left(\mu \right)u_{2}\left(\mu \right)d\mu \right\}, \end{array}$$
where
(79)$$\begin{array}{l} u_{1}\left(z\right)=\cos\left(z\cos\varphi -\gamma^{\prime }\right)+\sin\left(z\cos\varphi -\gamma^{\prime }-\varphi \right),\\ u_{2}\left(z\right)=\sin\left(z\cos\varphi -\gamma^{\prime }\right)-\cos\left(z\cos\varphi -\gamma^{\prime }-\varphi \right). \end{array}$$

Equation (78) supplies the shape of wall *C*’*E*’ in parametric form. In particular, the Cartesian coordinates of point *E*’ are determined from (78) as
(80)$$x_{E^{\prime }}=x_{C^{\prime }E^{\prime }}\left(\beta_{F}\right)\;\mathrm{and}\;y_{E^{\prime }}=y_{C^{\prime }E^{\prime }}\left(\beta_{F}\right).$$

### 4.4. Ideal Die Shapes

The general solution in Section 4.1, Section 4.2 and Section 4.3 determines the shapes of *DCE* and *D’C’E*’ (Figure 3) and contains three parameters: *γ*, *γ*’, and *â*. The process is classified by two independent parameters, *h*_in_/*h*_out_ and *ψ*_in_ (Figure 1). To calculate an ideal die for a specific process, one must connect the first group of the parameters and the second.

It is seen from Figure 3 that *ψ = ψ*_in_ at point *B*. On the other hand, *α* = *α_F’_*, and *β* = *β _F_*, at this point. Therefore, it follows from (9) and (26) that
(81)$$\psi_{\mathrm{in}}=\gamma -\gamma^{\prime }$$

In particular, *ψ*_in_ = 0, and *γ* = *γ*’, in the case of symmetric dies.

It follows from (10) and (11) that
(82)$$\ln\frac{V}{V_{0}}=\left(1-\sin\varphi \right)\left(\alpha -\beta \right).$$

Since *α* = 0 and *β* = 0 at point *A*, it follows from (82) that *V*_0_ = *V*_out_, and
(83)$$\ln\frac{V_{\mathrm{in}}}{V_{\mathrm{out}}}=\left(1-\sin\varphi \right)\left(\alpha_{F^{\prime }}-\beta_{F}\right).$$

The material is incompressible. Therefore, *V*_out_*h*_out_ = *V*_in_*h*_in_. This equation, (26), and (83) combine to give
(84)$$\ln\frac{h_{\mathrm{in}}}{h_{\mathrm{out}}}=\left(\gamma +\gamma^{\prime }\right)\cot\chi .$$

Solving Equations (81) and (84) for *γ* and *γ*’ supplies
(85)$$\gamma =\frac{1}{2}\left[\ln\left(\frac{h_{\mathrm{in}}}{h_{\mathrm{out}}}\right)\tan\chi +\psi_{\mathrm{in}}\right]\;\mathrm{are}\;\gamma^{\prime }=\frac{1}{2}\left[\ln\left(\frac{h_{\mathrm{in}}}{h_{\mathrm{out}}}\right)\tan\chi -\psi_{\mathrm{in}}\right].$$

Since *h*_in_/*h*_out_ > 1 and *ψ*_in_ ≥ 0, the first equation in (85) shows that the inequality in (20) is automatically satisfied. The second equation in (85) and the inequality in (25) impose the following restriction on the process parameters:(86)$$\ln\left(h_{\mathrm{in}}/h_{\mathrm{out}}\right)\tan\chi \geq \psi_{\mathrm{in}}.$$

The parameter *â* is not connected to *h*_in_/*h*_out_ and *ψ*_in_. It can be used for calculating different ideal dies for the same process (i.e., for the same values of *h*_in_/*h*_out_ and *ψ*_in_). This parameter may vary in the range
(87)$$-h_{\mathrm{out}}/2\leq \hat{a}\leq h_{\mathrm{out}}/2.$$

### 4.5. Special Solutions

Some regions shown in Figure 3 degenerate to lines or points at certain values of the parameters classifying the characteristic network. One of the special solutions is obtained if *γ*’ = 0. In this case, angle *AD*’*F*’ vanishes, and line *DAF*’*C*’ becomes straight. This straight line is the rigid/plastic boundary, and triangle *D*’*F*’*C*’ becomes a part of the rigid sheet that exits the die. Since line *AF*’ degenerates to a point, region *DAF*’*C*’*E*’*BF* becomes a characteristic fan with a point singularity *D*. Thus, line *DFBE*’ becomes straight, and region *DFBEC* becomes a triangle, which can be treated as a part of the rigid sheet that enters the die. Therefore, line *DFBE*’ is the rigid/plastic boundary. Figure 5 illustrates the characteristic network. It is seen from (85) that this special solution corresponds to the maximum possible value of *ψ*_in_ at a given ratio *h*_in_/*h*_out_. This maximum value is readily determined from (86). It is clear from the above that the value of *â* is immaterial for this special solution.

Another special solution is obtained if *â* = *h*_out_/2. Line *AD* and region *AFCD* degenerate to points. The solution given in Section 4.2 is not required for calculating the characteristic network. The value of *â* is immaterial for this special solution. Any value of *ψ*_in_ that satisfies (86) can be chosen. Figure 6 illustrates the characteristic network at *ψ*_in_ = 0.

## 5. Pressure on the Die’s Walls

The pressure on the die’s walls is important for the process design. In particular, it is involved in many empirical equations for calculating the wear of tools [30,31,32,33]. Since the die is frictionless, the normal stress acting on the die’s walls equals *σ_η_*. The principal stresses are readily found by employing the solution in Section 4.

In terms of the principal stresses, Equation (1) has the following form:(88)$$-p\sin\varphi +\frac{\sigma_{\xi }-\sigma_{\eta }}{2k}=\cos\varphi .$$

Using (8) and (88), one can represent the principal stresses as
(89)$$\sigma_{\xi }/k=\cos\varphi -\left(1-\sin\varphi \right)p\;\mathrm{and}\;\sigma_{\eta }/k=-\cos\varphi -\left(1+\sin\varphi \right)p.$$

Since *σ_ξ_* = 0 on lines *AD* and *AD*’, it follows from (10) and the first equation in (89) that
(90)$$p_{0}=\tan\chi .$$

Equations (12) and (90) supply the value of *p* in region *DFC*, where the principal stresses are constant. In particular,
(91)$$p_{DFC}=\left(\tan\chi +\cot\varphi \right)\exp\left(2\gamma \tan\varphi \right)-\cot\varphi .$$

Then, the dimensionless pressure on the die’s wall between points *D* and *C* is determined from (89) and (91) as
(92)$$q=\left|\sigma_{\eta }\right|/k=\left|-\cos\varphi -\left(1+\sin\varphi \right)\left[\left(\tan\chi +\cot\varphi \right)\exp\left(2\gamma \tan\varphi \right)-\cot\varphi \right]\right|.$$

Similarly, the dimensionless pressure on the die’s wall between points *D*’ and *C*’ is determined as
(93)$$q=\left|\sigma_{\eta }\right|/k=\left|-\cos\varphi -\left(1+\sin\varphi \right)\left[\left(\tan\chi +\cot\varphi \right)\exp\left(2\gamma^{\prime }\tan\varphi \right)-\cot\varphi \right]\right|.$$

Note that
(94)$$\underset{\varphi \to 0}{\lim}q=2+2\gamma \;\mathrm{and}\;\underset{\varphi \to 0}{\lim}q=2+2\gamma^{\prime }$$
on walls *DC* and *D*’*C*’, respectively.

Equation (13) is valid in region *CFBE*. Therefore, taking into account (90), one can get
(95)p=(tanχ+cotφ)exp[2(γtanφ−αsinφ)]−cotφ.

The second equation in (89) and (95) combine to give
(96)q=|−cosφ−(1+sinφ){(tanχ+cotφ)exp[2(γtanφ−αsinφ)]−cotφ}|.

This equation provides the pressure on the die between points *C* and *E*. Equations (65) and (96) allow the pressure distribution to be calculated in the Cartesian coordinates.

One can similarly derive the equation for the pressure on the die between points *C*’ and *E*’. As a result,
(97)$$q=\left|-\cos\varphi -\left(1+\sin\varphi \right)\left\{\left(\tan\chi +\cot\varphi \right)\exp\left[2\left(\gamma^{\prime }\tan\varphi +\beta \sin\varphi \right)\right]-\cot\varphi \right\}\right|.$$

This equation and (78) allow the pressure distribution to be calculated in the Cartesian coordinates.

It follows from (7) that *p* is constant along lines *EB* and *E’B*. Therefore, it is constant along the entire rigid/plastic boundary. Thus, the value of *p* is the same at points *E* and *E*’. Then, it follows from (89) that the value of *q* is the same at these points. This value can be found from (96) or (97) using (26). As a result,
(98)$$q_{E,E^{\prime }}=\left|-\cos\varphi -\left(1+\sin\varphi \right)\left\{\left(\tan\chi +\cot\varphi \right)\exp\left[2\left(\gamma +\gamma^{\prime }\right)\tan\varphi \right]-\cot\varphi \right\}\right|.$$

Eliminating in this equation *γ* and *γ*’ employing (84), one gets
(99)$$q_{E,E^{\prime }}=\left|-\cos\varphi -\left(1+\sin\varphi \right)\left\{\left(\tan\chi +\cot\varphi \right){\left(h_{\mathrm{in}}/h_{\mathrm{out}}\right)}^{2\tan\varphi \tan\chi }-\cot\varphi \right\}\right|.$$

It follows from this equation that
(100)$$\underset{\varphi \to 0}{\lim}q_{E,E^{\prime }}=2+2\left(\gamma +\gamma^{\prime }\right)=2+2\ln\left(h_{\mathrm{in}}/h_{\mathrm{out}}\right)$$

Equation (99) shows that the value of *q* at points *E* and *E*’ is independent of *ψ*_in_.

Since the value of *q* on the rigid/plastic boundary has been found, the principal stress $\sigma_{\xi }$ is readily determined from (89) by eliminating *p*. The quantity $\sigma_{\xi }/k$ may be regarded as a dimensionless extrusion force.

## 6. Illustrative Examples

This section presents the numerical results demonstrating the process parameters’ effect on the ideal die shape and wall pressure. The deviation from the symmetric shape is emphasized. All the calculations have been performed for *h*_in_/*h*_out_ = 2. Figure 7, Figure 8 and Figure 9 show several die profiles and the corresponding wall pressure distributions at *â* =0. The parameter *ψ*_in_ varies from 0 (symmetric dies) to its maximum value determined by Equation (86). Different figures correspond to different values of the angle of internal friction. In particular, *φ* = 0 in Figure 7 (pressure-independent material); *φ* = 15° in Figure 8; and *φ* = 30° in Figure 9.

The influence of parameter *â* is revealed in Figure 10, Figure 11, Figure 12 and Figure 13. In these cases, *φ* = 0. As in Figure 7, Figure 8 and Figure 9 the parameter *ψ*_in_ varies from 0 to its maximum value. Also, *â* = *h*_out_/4 in Figure 10; *â* = *h*_out_/2 in Figure 11; *â* = −*h*_out_/4 in Figure 12; and *â* = −*h*_out_/2 in Figure 13. The dies in Figure 11 and Figure 13 correspond to the special solutions discussed in Section 4.5.

The numerical results show that the length of the upper wall decreases, and the length of the lower wall increases as parameter *ψ*_in_ increases. The leftmost point of the upper die is to the right of the leftmost point of the lower die if *â* > 0. Accordingly, the leftmost point of the upper die is to the left to the leftmost point of the lower die if *â* < 0. The pressure on the upper die is always higher than on the lower die, except for the symmetric dies.

## 7. Conclusions

A new general stationary planar ideal flow solution has been found for the double slip and rotation model. Its distinguishing feature, as compared to other planar ideal flow solutions, is that the flow has no symmetry axis. The solution in Section 4 includes all possible ideal flows, assuming that the characteristic field is singular at the die’s exit (Figure 3). Generally, the characteristic network consists of three types of regions. Both families of characteristics are straight in the regions of one type; the characteristics of one family are curved in the regions of the second type; and both families of characteristics are curved in the regions of the third type. However, some of these regions may not exist under certain conditions. The solution is purely analytical in the regions where the characteristics of at least one family are curved. Riemann’s method is employed for calculating the characteristic network in the regions where both families of characteristics are curved. In this case, a numerical technique is only necessary to evaluate ordinary integrals. It is known that Riemann’s method is very accurate, and corresponding solutions can be used to verify the accuracy of numerical solutions [34]. The general solution is used to show the effect of process and material parameters on the ideal die’s shape and the distribution of wall pressures. The corresponding symmetric shapes are obtained as a particular case. The die’s shapes calculated are compared to shapes for pressure-independent material.

## Figures and Tables

**Figure 1 materials-16-07378-f001:**
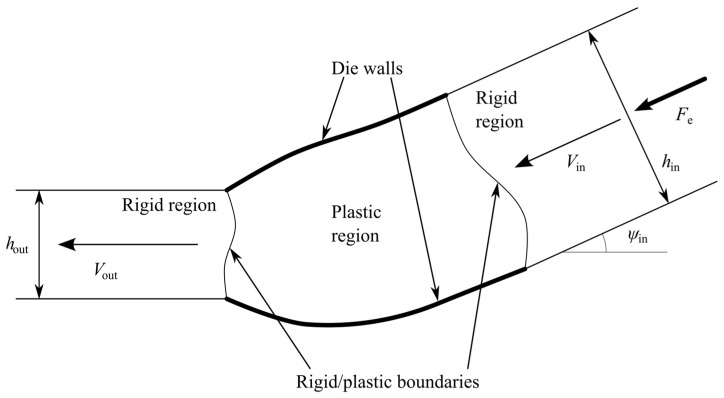
Schematic diagram of the extrusion process.

**Figure 2 materials-16-07378-f002:**
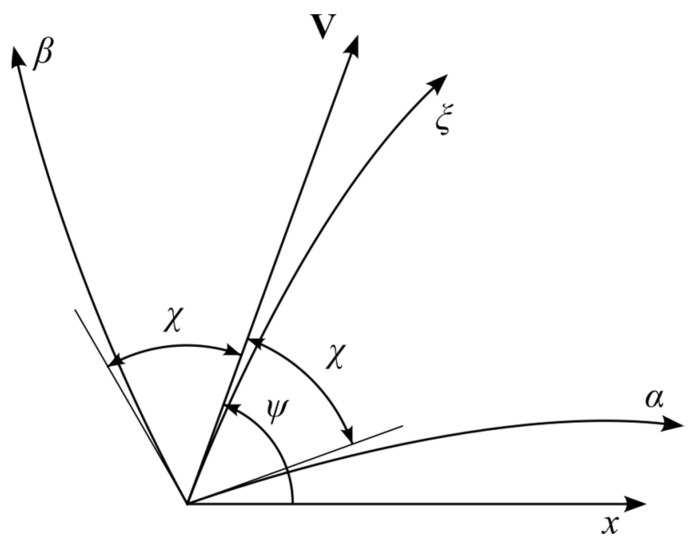
Characteristic coordinates and the velocity vector.

**Figure 3 materials-16-07378-f003:**
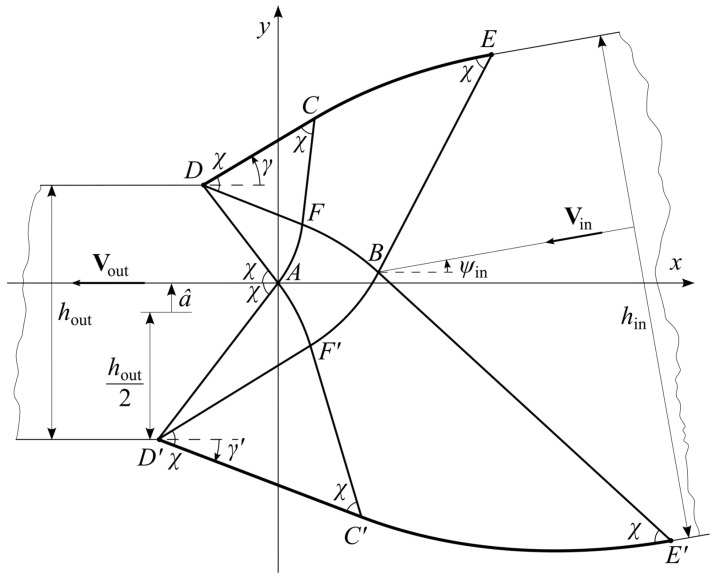
General structure of the characteristic network.

**Figure 4 materials-16-07378-f004:**
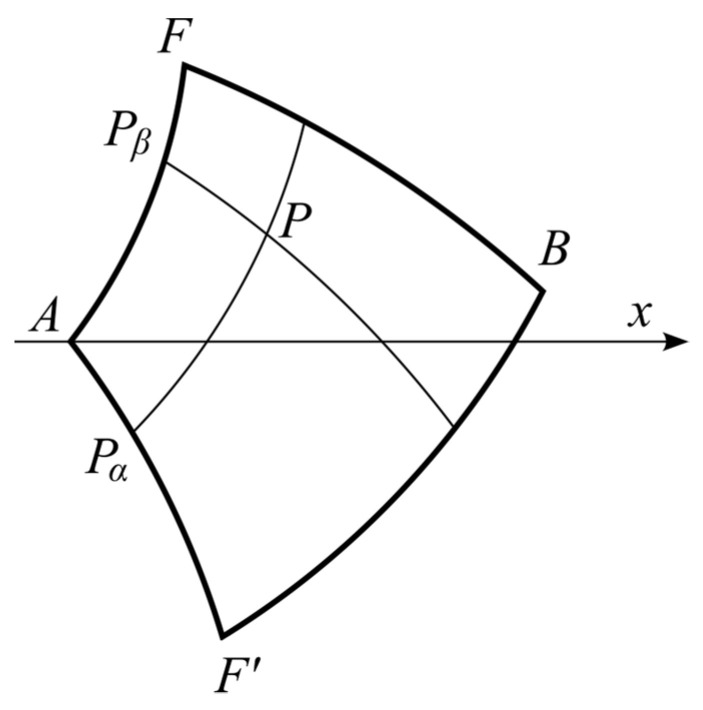
Characteristic lines for Riemann’s method of integration (*α* = *a* on *PP_α_*, and *β* = *b* on *PP_β_*).

**Figure 5 materials-16-07378-f005:**
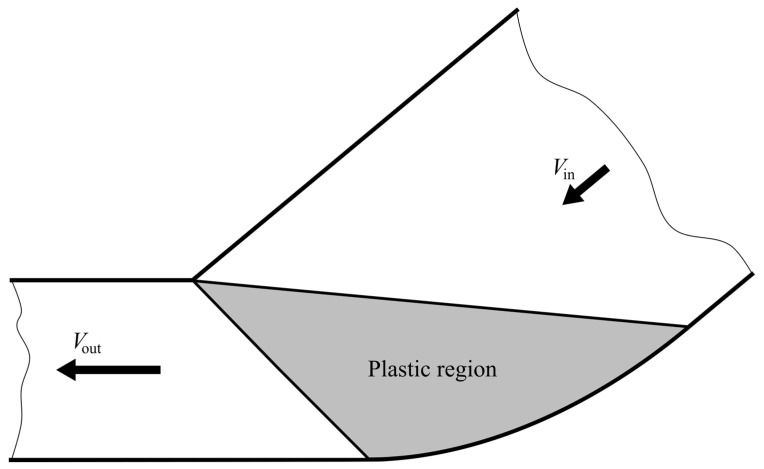
General structure of the characteristic network corresponding to the special solution *γ*’ = 0.

**Figure 6 materials-16-07378-f006:**
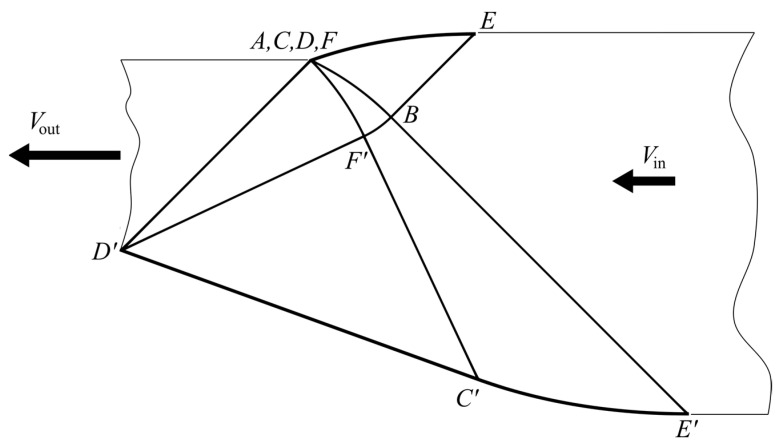
General structure of the characteristic network corresponding to the special solution *â* = *h*_out_/2 at *ψ*_in_ = 0.

**Figure 7 materials-16-07378-f007:**
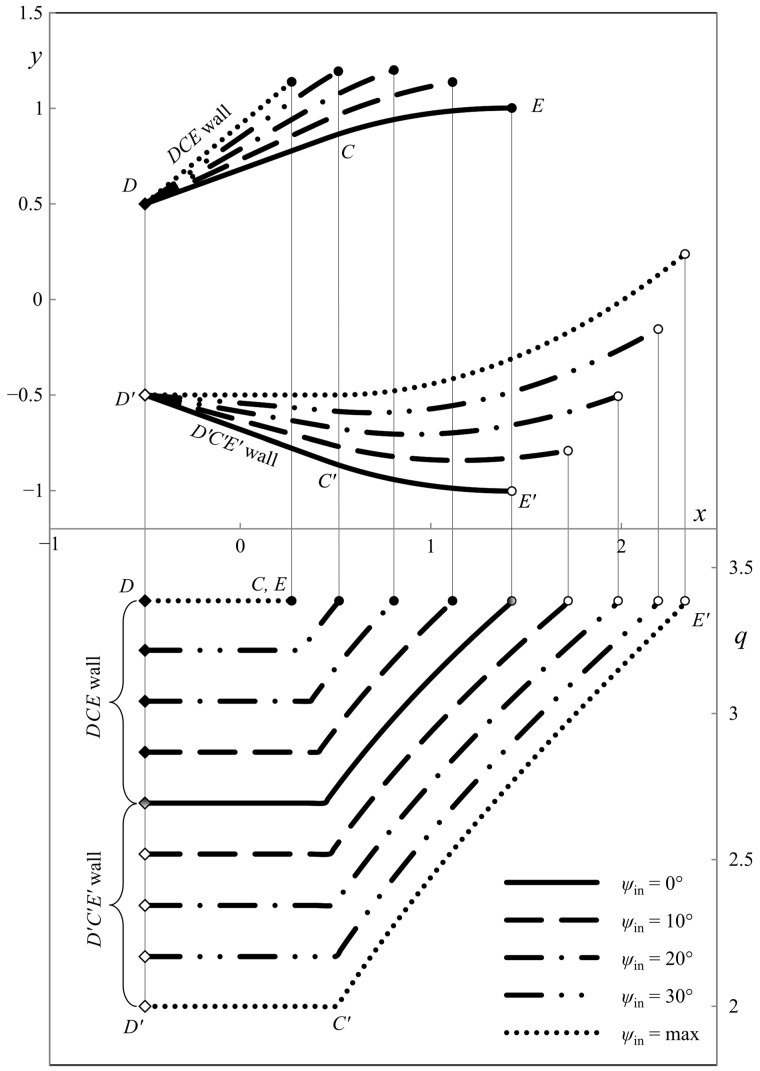
Die shape (upper diagram) and wall pressures (lower diagram) at *φ* = 0° and *â* = 0.

**Figure 8 materials-16-07378-f008:**
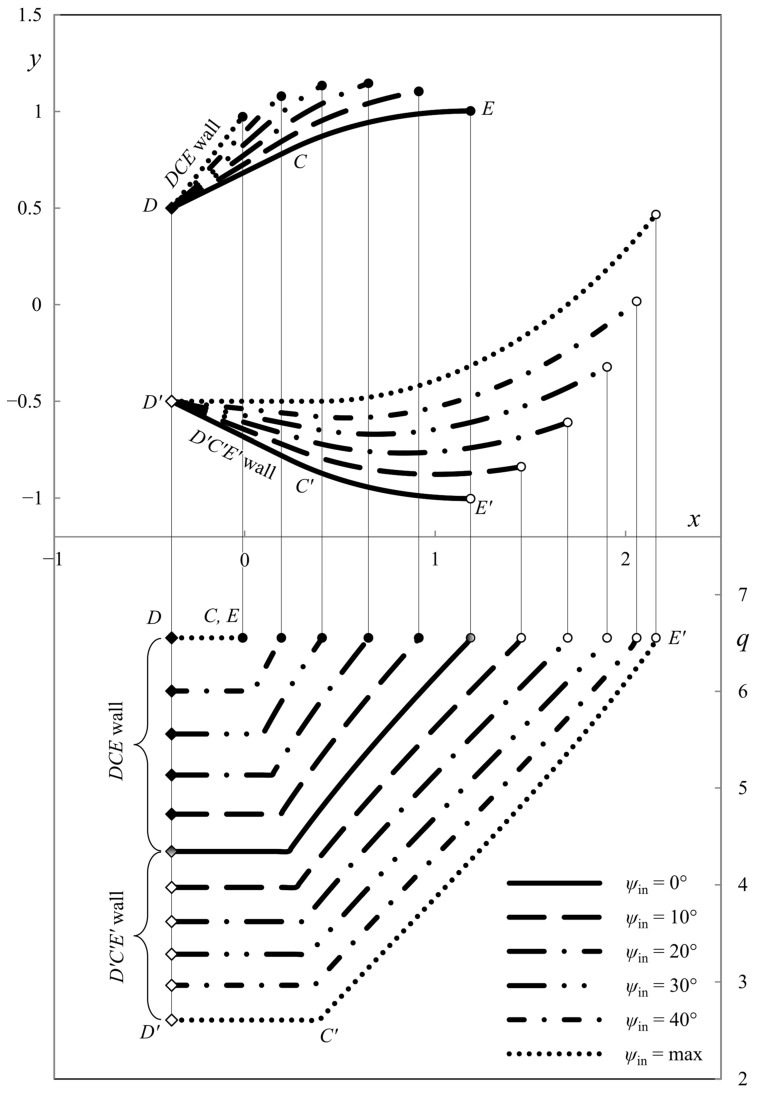
Die shape (upper diagram) and wall pressures (lower diagram) at *φ* = 15° and *â* = 0.

**Figure 9 materials-16-07378-f009:**
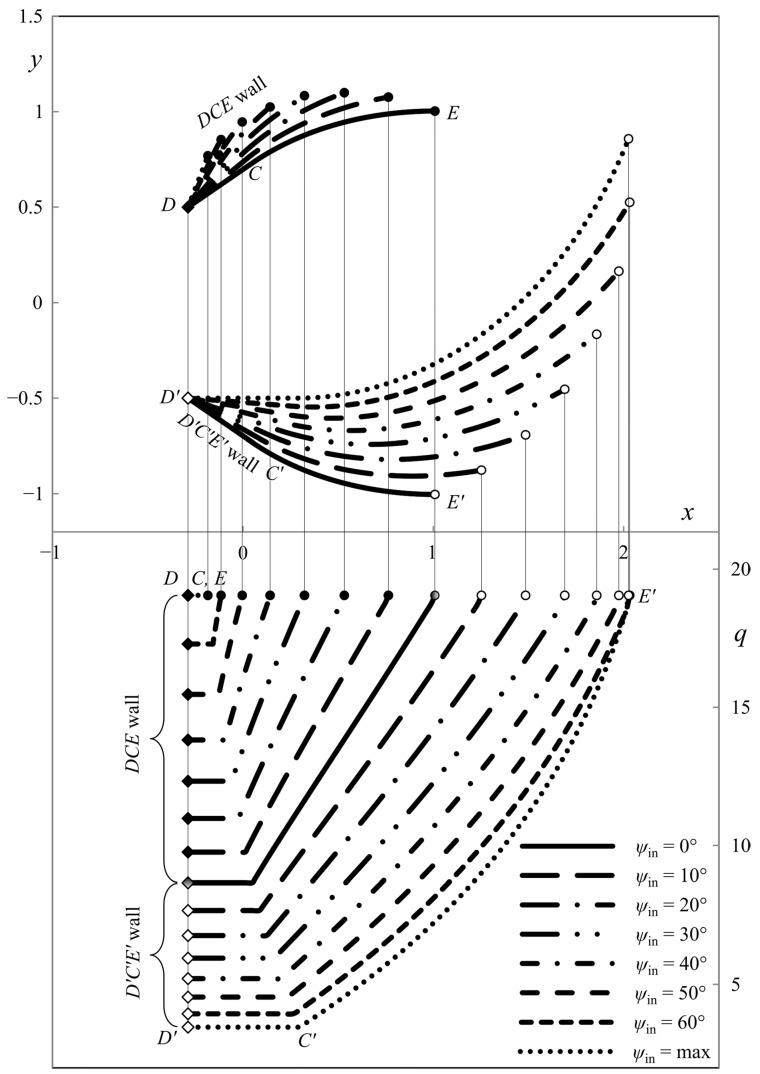
Die shape (upper diagram) and wall pressures (lower diagram) at *φ* = 30° and *â* = 0.

**Figure 10 materials-16-07378-f010:**
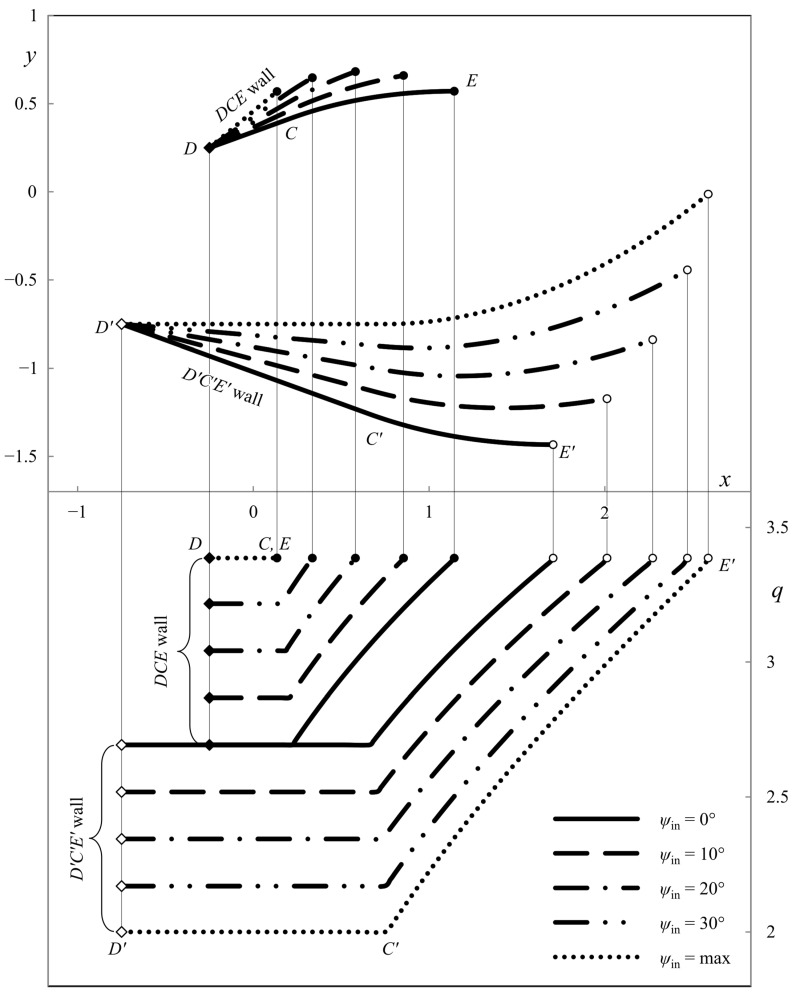
Die shape (upper diagram) and wall pressures (lower diagram) at *φ* = 0° and *â* = *h*_out_/4.

**Figure 11 materials-16-07378-f011:**
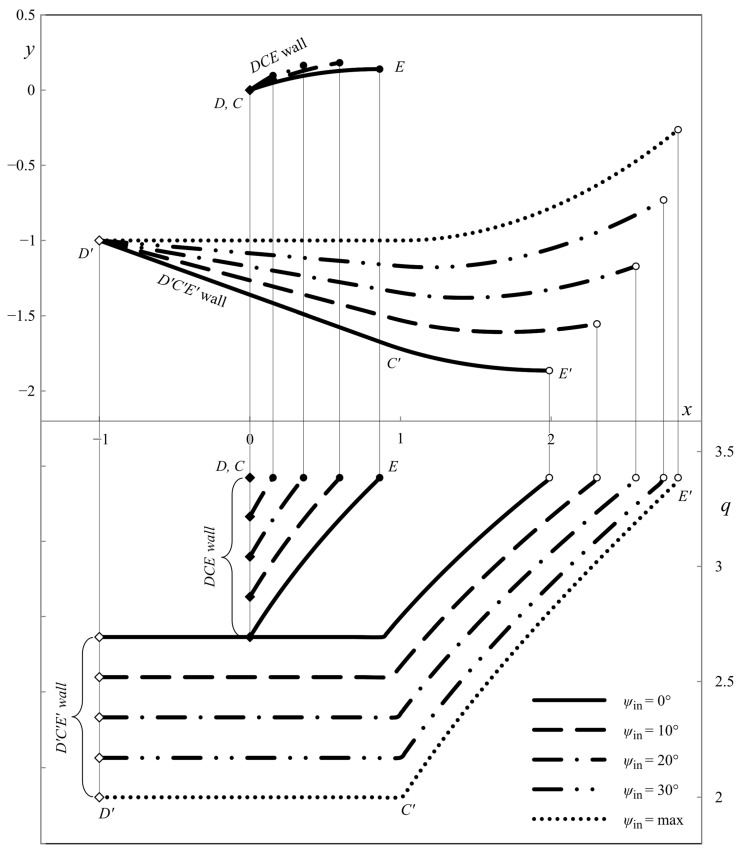
Die shape (upper diagram) and wall pressures (lower diagram) at *φ* = 0° and *â* = *h*_out_/2.

**Figure 12 materials-16-07378-f012:**
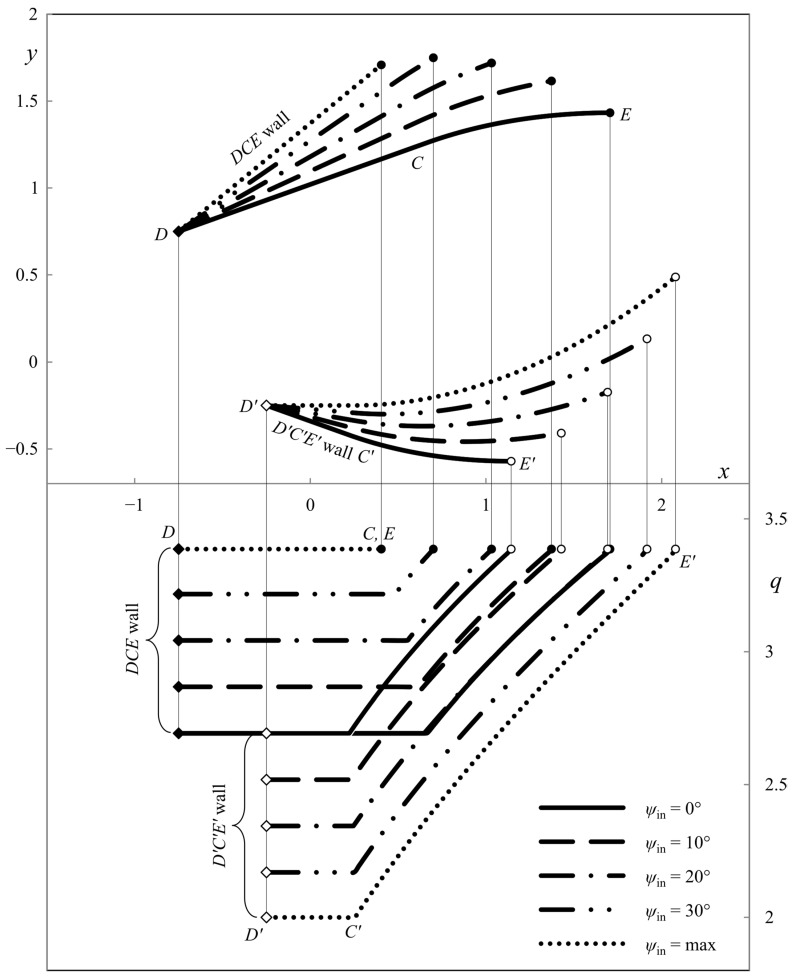
Die shape (upper diagram) and wall pressures (lower diagram) at *φ* = 0° and *â* = −*h*_out_/4.

**Figure 13 materials-16-07378-f013:**
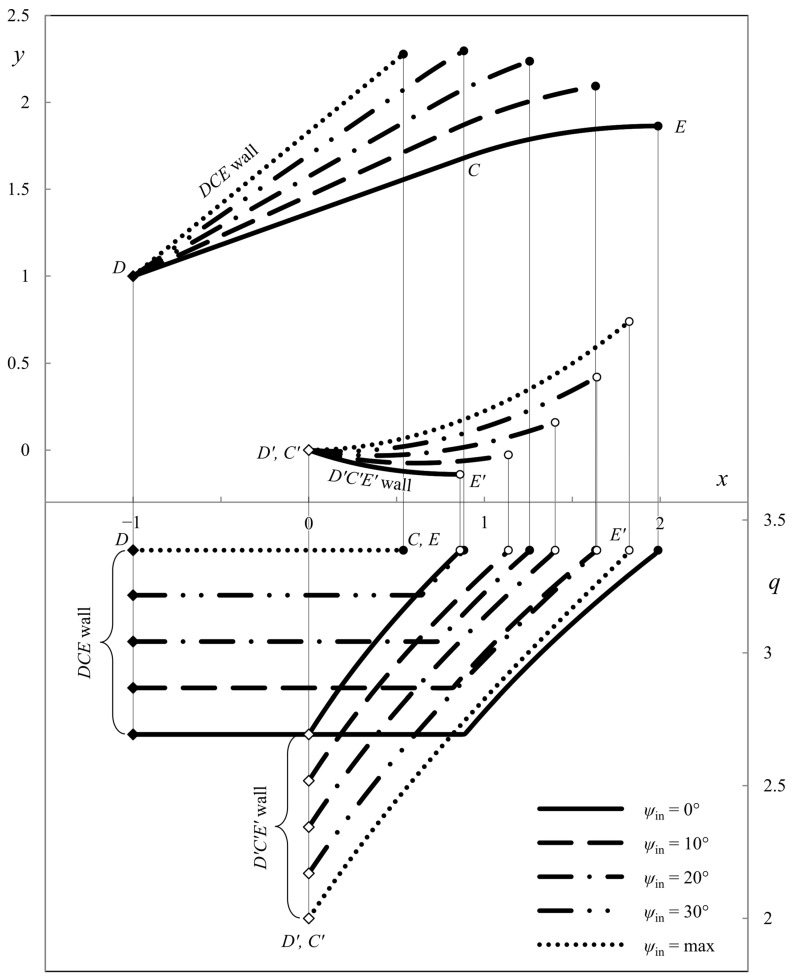
Die shape (upper diagram) and wall pressures (lower diagram) at *φ* = 0° and *â* = −*h*_out_/2.

## Data Availability

Data is contained within the article.

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
