# Peer review of "General Planar Ideal Flow Solutions with No Symmetry Axis"

_materials, 2023, doi:10.3390/ma16237378_

Round 1

Reviewer 1 Report

Comments and Suggestions for Authors

Re: General planar  ideal flow solutions with no symmetry axis.

S.     Alexandrov, V. Mokryakov

The paper is addressed to analysis of  plane plastic flow of pressure sensitive materials, satisfying Mohr-Coulomb yield condition and executing a so called ideal flow mode with flow velocity oriented along the diagonal direction relative to slip lines. Such flow mode was analyzed much earlier by Richmond and Morrison, Ref. (3), who provided the slip line solution for the extrusion problem for metals and specified the die shape as  a part of solution. The present paper belongs to a  series of publications on this topic. It was extended by the authors to materials for which the yield condition depends on hydrostatic pressure, Refs. (11, 14). Regrettably, the presentation is not clear for a reader not specialized in the methodology of analysis, which requires study of literature papers.

1.The paper title should be changed, for instance as: Analysis of plane plastic flow in non-symmetric extrusion  of pressure sensitive materials.

2. Introduction should provide clear explanation of the used methodology and the concept of ideal flow mode explaining why such flow provides improved solution, also specifying the class of pressure sensitive materials for which the presented analysis can be applied. The constitutive relations should also be clearly discussed. In the paper it is mentioned that the double sliding model combined with element rotation,  as discussed in Harris et al. paper, Ref. (10), is used. This certainly refers to  granular  materials, soils and powders. But for these materials the critical state models accounting for dilatancy, compaction and critical states are most appropriate. However, the assumed model         is based on the assumption of isochoric deformation not allowing for material compaction in the  extrusion process. The analysis presented seems highly unrealistic in such cases. The neglect of friction effect at material-tool interfaces makes the solution inapplicable to real processes. For metals Mohr-Coulomb condition   usually cannot be applied as pressure effect on the yield stress is negligible. The observed yield stress difference in tension and compression results from the effect of third stress deviator invariant.

3. Section 2 containing statement of the problem is simply copied from Ref. (11) of the same authors with the same formulas and figures. The solution of the symmetric extrusion was presented in (11).  

4. The application of analysis to some specific cases should be presented , demonstrating   loading dependence on the cross section reduction and flow mode, also relation to material parameters such as cohesion and friction values. What advantage provides the ideal flow  solution and a solution  for straight die profile?

 The reviewer is doubtful about publication of  paper in the present form. It needs essential improvement.

Comments on the Quality of English Language

Re: General planar  ideal flow solutions with no symmetry axis.

S.     Alexandrov, V. Mokryakov

The paper is addressed to analysis of  plane plastic flow of pressure sensitive materials, satisfying Mohr-Coulomb yield condition and executing a so called ideal flow mode with flow velocity oriented along the diagonal direction relative to slip lines. Such flow mode was analyzed much earlier by Richmond and Morrison, Ref. (3), who provided the slip line solution for the extrusion problem for metals and specified the die shape as  a part of solution. The present paper belongs to a  series of publications on this topic. It was extended by the authors to materials for which the yield condition depends on hydrostatic pressure, Refs. (11, 14). Regrettably, the presentation is not clear for a reader not specialized in the methodology of analysis, which requires study of literature papers.

1.The paper title should be changed, for instance as: Analysis of plane plastic flow in non-symmetric extrusion  of pressure sensitive materials.

2. Introduction should provide clear explanation of the used methodology and the concept of ideal flow mode explaining why such flow provides improved solution, also specifying the class of pressure sensitive materials for which the presented analysis can be applied. The constitutive relations should also be clearly discussed. In the paper it is mentioned that the double sliding model combined with element rotation,  as discussed in Harris et al. paper, Ref. (10), is used. This certainly refers to  granular  materials, soils and powders. But for these materials the critical state models accounting for dilatancy, compaction and critical states are most appropriate. However, the assumed model                                                                                                    is based on the assumption of isochoric deformation not allowing for material compaction in the  extrusion process. The analysis presented seems highly unrealistic in such cases. The neglect of friction effect at material-tool interfaces makes the solution inapplicable to real processes. For metals Mohr-Coulomb condition   usually cannot be applied as pressure effect on the yield stress is negligible. The observed yield stress difference in tension and compression results from the effect of third stress deviator invariant.

3. Section 2 containing statement of the problem is simply copied from Ref. (11) of the same authors with the same formulas and figures. The solution of the symmetric extrusion was presented in (11).  

4. The application of analysis to some specific cases should be presented , demonstrating   loading dependence on the cross section reduction and flow mode, also relation to material parameters such as cohesion and friction values. What advantage provides the ideal flow  solution and a solution  for straight die profile?

 The reviewer is doubtful about publication of  paper in the present form. It needs essential improvement.

Author Response

Dear Reviewer:

Our answers to your comments are provided below. The changes made in the manuscript are marked in red. The references follow the revised manuscript.

The paper is addressed to analysis of plane plastic flow of pressure sensitive materials, satisfying Mohr-Coulomb yield condition and executing a so called ideal flow mode with flow velocity oriented along the diagonal direction relative to slip lines. Such flow mode was analyzed much earlier by Richmond and Morrison, Ref. (3) ([4] in the revised manuscript), who provided the slip line solution for the extrusion problem for metals and specified the die shape as a part of solution. The present paper belongs to a series of publications on this topic. It was extended by the authors to materials for which the yield condition depends on hydrostatic pressure, Refs. (11, 14) ([19] and [22] in the revised manuscript). Regrettably, the presentation is not clear for a reader not specialized in the methodology of analysis, which requires study of literature papers.

Response.

We agree that Owen Richmond and his co-workers initiated the ideal flow theory in the 60th. However, the first paper was published in 1962 ([2] in the list of references to our paper). The paper you noted extended this first result to axisymmetric flows. We have extended the Introduction to provide more information on the methodology.

  1. The paper title should be changed, for instance as: Analysis of plane plastic flow in non-symmetric extrusion of pressure sensitive materials.

Response.

The proposed title means a problem has been solved under standard boundary conditions. In particular, the tool’s shape is given. It is not so in our paper. As you noted above, the tool’s shape is part of ideal flow solutions. The shapes shown in Section 6 are for illustration of the general solution. The general solution provides all possible shapes for producing planar ideal flows, assuming the singular characteristic field at the die’s exit. We have emphasized it in Section 7 of the revised manuscript.

  1. Introduction should provide clear explanation of the used methodology and the concept of ideal flow mode explaining why such flow provides improved solution, also specifying the class of pressure sensitive materials for which the presented analysis can be applied. The constitutive relations should also be clearly discussed. In the paper it is mentioned that the double sliding model combined with element rotation, as discussed in Harris et al. paper, Ref. (10) ([18] in the revised manuscript), is used. This certainly refers to granular materials, soils and powders. But for these materials the critical state models accounting for dilatancy, compaction and critical states are most appropriate. However, the assumed model is based on the assumption of isochoric deformation not allowing for material compaction in the extrusion process. The analysis presented seems highly unrealistic in such cases. The neglect of friction effect at material-tool interfaces makes the solution inapplicable to real processes. For metals Mohr-Coulomb condition usually cannot be applied as pressure effect on the yield stress is negligible. The observed yield stress difference in tension and compression results from the effect of third stress deviator invariant.

Response.

Section 2 explains the constitutive relations. However, you criticize this section below. As to all other issues in this comment, we do not propose this theory (we use the theory available in the literature), and we do not propose the material model (we use the model available in the literature). The novelty and significance of our findings are presented in Sections 4 to 6.

We have extended the Introduction to show that the material model we use (i.e., plastically incompressible material obeying the Mohr-Coulomb yield criterion) is used for many materials (of course, there are many materials for which this model is not acceptable). We have also emphasized the role of the ideal flow theory in the deformation process design. It is always used for the preliminary design. The limitations of the theory you noted are not specific to our solutions. The tool must be frictionless to generate any stationary ideal flow. Nevertheless, the theory is used in industrial applications.

  1. Section 2 containing statement of the problem is simply copied from Ref. (11) ([19] in the revised manuscript), of the same authors with the same formulas and figures. The solution of the symmetric extrusion was presented in (11) ([19] in the revised manuscript).

Response.

We agree. However, we must provide the constitutive equations and introduce the notation. This section contains one figure. There is no similar figure in [19].

  1. The application of analysis to some specific cases should be presented, demonstrating loading dependence on the cross section reduction and flow mode, also relation to material parameters such as cohesion and friction values. What advantage provides the ideal flow solution and a solution for straight die profile?

Response.

Section 5 provides much more information than you have requested. It provides the stress distribution along the contact surfaces. The rightmost points in the figures that illustrate the pressure distribution correspond to the die’s entry. Since the rigid/plastic boundaries are straight, the principal stresses are constant, as follows from (7). It is explained in detail at the end of Section 5. We have added a further explanation of how to calculate the extrusion force. The dimensionless extrusion force cannot depend on the cohesion. It follows from dimensional analysis.

An advantage of ideal flows is that these processes produce no redundant work. Of course, this work does not vanish in real experiments, but it is small. We have referred to an experimental paper (Reference [3]) to support this statement.

Reviewer 2 Report

Comments and Suggestions for Authors

Dear Authors: General planar ideal flow solutions with no symmetry axis”.  The article presents research on ideal flow solutions with no symmetry axis. Based on this theory, optimal extruders and minimum length dies are calculated. The development of the proposed theory is well described and discussed. The manuscript has an orderly structure. In my opinion, the article requires minor improvement and supplementation. Comments on the editorial page of the work that I think should be corrected:

A.      „Abstract”. I suggest emphasizing the purpose of the work and highlighting its main achievement. B.      „Keywords” In my opinion, it is better to separate the double sliding and rotation model into a double sliding, rotation model. C.          “Introduction” I know that the work is quite extensive, but in my opinion the introduction should be supplemented and the state of the literature should be clearly emphasized. Please also emphasize the importance of the considerations presented in the article. D.         „Statement of the Problem” I wanted to ask the authors whether the scheme presented in Fig. 1 was not based on literature premises or similar previous considerations of their own? If so, it would be good to mark it. E.          In my opinion, the mathematical description presented in the work is correct and interesting. These theories are important when developing new solutions that find practical application. I would add in the work the anticipated or already implemented application of the developed solution. F.          Figures 7-13 included in the work should be included in the calculation results chapter, not in the conclusion. I believe that they should be placed in the work in the places where they are described.

Author Response

Dear Reviewer,

Our answers to your comments are provided below. The changes made in the manuscript are marked in red. The references follow the revised manuscript.

Dear Authors: “General planar ideal flow solutions with no symmetry axis”.  The article presents research on ideal flow solutions with no symmetry axis. Based on this theory, optimal extruders and minimum length dies are calculated. The development of the proposed theory is well described and discussed. The manuscript has an orderly structure. In my opinion, the article requires minor improvement and supplementation. Comments on the editorial page of the work that I think should be corrected:

A. „Abstract”.I suggest emphasizing the purpose of the work and highlighting its main achievement. 

We have corrected the abstract.

B. „Keywords” In my opinion, it is better to separate the double sliding and rotation model into a double sliding, rotation model. 

We have not proposed this term. The model’s author (Prof. David Harris) named it the double sliding and rotation model.

C. “Introduction”I know that the work is quite extensive, but in my opinion the introduction should be supplemented and the state of the literature should be clearly emphasized. Please also emphasize the importance of the considerations presented in the article.

We have extended the Introduction.

D. „Statement of the Problem” I wanted to ask the authors whether the scheme presented in Fig. 1 was not based on literature premises or similar previous considerations of their own? If so, it would be good to mark it. 

Figure 1 is a schematic diagram. It is a graphical representation of the common knowledge about a class of rigid plastic solutions. We need this figure to introduce notation.

E. In my opinion, the mathematical description presented in the work is correct and interesting. These theories are important when developing new solutions that find practical application. I would add in the work the anticipated or already implemented application of the developed solution.

The main result is die’s shapes. Because these shapes have been calculated in the present paper, nobody could implement the solution for applications. We have added a reference to an experimental paper (Paper [3]). The results reported in this paper demonstrate how ideal flow solutions can be used in practice.

F. Figures 7-13 included in the work should be included in the calculation results chapter, not in the conclusion. I believe that they should be placed in the work in the places where they are described.

We have moved the figures as suggested.

Reviewer 3 Report

Comments and Suggestions for Authors

The topic is interesting and would be of interest to the readers of  Materials. The findings are also good, and the paper makes an acceptable contribution. However, the results need to be validated by conducting numerical simulations or experimental tests

Author Response

Dear Reviewer:

Our answers to your comments are provided below. The changes made in the manuscript are marked in red. The references follow the revised manuscript.

The topic is interesting and would be of interest to the readers of  Materials. The findings are also good, and the paper makes an acceptable contribution. However, the results need to be validated by conducting numerical simulations or experimental tests

Response.

We have added a reference to an experimental paper (Paper [3]). The results reported in this paper demonstrate how ideal flow solutions can generally be used in practice. The new shapes calculated in our paper cannot be verified before publication because nobody knows them.

The finite-difference method in characteristic coordinates is the best numerical method for solving inverse problems of the type considered in our paper. We guess that the reviewer means this method. However, Riemann’s method is more accurate than the finite-difference method. It has been confirmed in the literature. We have referred to [30], where the comparison was made. The paper concluded that solutions based on Riemann’s method can be used to verify numerical solutions.

The traditional finite element method cannot calculate ideal flows. We have added this remark to the Introduction.

Round 2

Reviewer 1 Report

Comments and Suggestions for Authors

The analysis is based assuming pressure dependent Mohr-Coulomb yield condition and incompressibility of plastic flow. This assumption corresponds to the non-associted flow rule. Then classical plasticity theory predicts orthogonal network of flow characteristics., differing from stress state characteristics. However, for double shearing model (as proposed by Spencer) and used in this paper, a differing flow rule applies involving stress and strain rate relation. It is questionable whether such model can be used for metals. For soils, granulates or powders, the extrusion process is usually related to strong compaction with density increase, contrary to the incompressibility assumption.  It is difficult to assess  applicabilty of the presented die design to real materials. In the paper a more detailed discussion  on model validity should be presented.

Comments on the Quality of English Language

English text requires some improvements

Author Response

Dear Reviewer:

Comments and Suggestions for Authors

The analysis is based assuming pressure dependent Mohr-Coulomb yield condition and incompressibility of plastic flow. This assumption corresponds to the non-associted flow rule. Then classical plasticity theory predicts orthogonal network of flow characteristics., differing from stress state characteristics. However, for double shearing model (as proposed by Spencer) and used in this paper, a differing flow rule applies involving stress and strain rate relation. It is questionable whether such model can be used for metals. For soils, granulates or powders, the extrusion process is usually related to strong compaction with density increase, contrary to the incompressibility assumption.  It is difficult to assess  applicabilty of the presented die design to real materials. In the paper a more detailed discussion  on model validity should be presented.

Response.

The ideal flow condition is incompatible with Spencer’s model. For this reason, the corresponding reference was not included in the original manuscript. We included it in the previous revised version to address your request to show the applicability of plastically incompressible models. We have clarified in the new version of the manuscript that the ideal flow condition is incompatible with Spencer’s model.

Concerning your main critical comment, in the previous revised version, we included several publications, including experimental work, that confirm the applicability of plastically incompressible models for various materials. The new version of the manuscript includes one more reference to experimental work on granular materials. Moreover, we have included a reference to a paper devoted to micromechanical modeling of granular materials. This paper concludes that the model we use is acceptable for some materials. We have also emphasized that many materials require plastically compressible models, as stated in your comment.

 Comments on the Quality of English Language

English text requires some improvements

Response.

The premium version of Grammarly has detected no grammatical errors. Of course, the style may not be as good as that of a text written by an English-speaking author. Nevertheless, we have asked our English-speaking colleague to read the manuscript. He has not found obvious problems with the language. Finally, a MDPI department checks the usage of English, and we do not doubt that they will ask us to correct it if necessary. Also, we will be happy to incorporate any specific comments from you.
